# Optimal Scaling for Locally Balanced Proposals in Discrete Spaces

**Haoran Sun**[*]
Georgia Tech
hsun349@gatech.edu

**Hanjun Dai**
Google Brain
hadai@google.com

**Dale Schuurmans**
Google Brain, U of Alberta
schuurmans@google.com

## Abstract

Optimal scaling has been well studied for Metropolis-Hastings (M-H) algorithms in continuous spaces, but a similar understanding has been lacking in discrete spaces. Recently, a family of locally balanced proposals (LBP) for discrete spaces has been proved to be asymptotically optimal, but the question of optimal scaling has remained open. In this paper, we establish, for the first time, that the efficiency of M-H in discrete spaces can also be characterized by an asymptotic acceptance rate that is independent of the target distribution. Moreover, we verify, both theoretically and empirically, that the optimal acceptance rates for LBP and random walk Metropolis (RWM) are $0.574$ and $0.234$ respectively. These results also help establish that LBP is asymptotically $O(N^{\frac{2}{3}})$ more efficient than RWM with respect to model dimension $N$. Knowledge of the optimal acceptance rate allows one to automatically tune the neighborhood size of a proposal distribution in a discrete space, directly analogous to step-size control in continuous spaces. We demonstrate empirically that such adaptive M-H sampling can robustly improve sampling in a variety of target distributions in discrete spaces, including training deep energy based models.

## 1 Introduction

The Markov Chain Monte Carlo (MCMC) algorithm is one of the most widely used methods for sampling from intractable distributions (Robert & Casella, 2013). An important class of MCMC algorithms is Metropolis-Hastings (M-H) (Metropolis et al., 1953; Hastings, 1970), where new states are generated from a proposal distribution followed by a M-H test. The efficiency for M-H algorithms depends critically on the proposal distribution. For example, gradient based methods, such as the Metropolis Adjusted Langevin Algorithm (MALA) (Rossky et al., 1978), Hamiltonian Monte Carlo (HMC) (Neal et al., 2011), and their variants Girolami & Calderhead (2011); Hoffman et al. (2014) substantially improve the performance of M-H algorithms in theory and in practice, compared to naive Random Walk Metropolis (RWM), by leveraging gradient information to guide the proposal distribution (Roberts & Rosenthal, 2001).

Despite many advances, progress in gradient based methods has generally focused on continuous spaces. However, Zanella (2020) recently proposed a general framework of locally balanced proposals (LBP) for discrete spaces, where a proposal distribution is designed to utilize probability changes between states. Subsequently, Grathwohl et al. (2021) accelerated the sampler by using gradient information to approximate the probability change. In empirical evaluations, similar to gradient based samplers in continuous spaces, LBP significantly outperforms RWM and other samplers in discrete spaces. However, both Zanella (2020) and Grathwohl et al. (2021) constrain the proposal distribution to lie within a 1-Hamming ball; i.e., only one site of the state variable is allowed to change per M-H

---

[*]Work done during an internship at Google.

step. Such a restricted update reduces the efficiency of the sampler. Sun et al. (2021) noticed this problem and modified the proposal distribution to allow multiple sites to be changed per M-H step. Although such larger updates significantly improve efficiency, Sun et al. (2021) do not show how to determine the update size, leaving the number of sites updated in an M-H step as a hyperparameter to tune.

In continuous spaces, the scale of the proposal distribution is known to be a critical hyperparameter for obtaining an efficient M-H sampler. For example, consider a Gaussian proposal $\mathcal{N}(x, \sigma^2)$ for modifying a current state $x$ with scale $\sigma$. If $\sigma$ is too small, the Markov chain will converge slowly since its increments will be small. Conversely, if $\sigma$ is too large, the M-H test will reject too high a proportion of proposed updates. A significant literature has studied optimal scaling for gradient based methods in continuous spaces (Gelman et al., 1997; Roberts & Rosenthal, 1998, 2001; Beskos et al., 2013), showing that the optimal scaling can be adaptively tuned w.r.t. the acceptance rate, independent of the target distribution. Such results suggest a direction for solving the optimal scaling problem for LBP. However, the underlying techniques for approximating a diffusion process cannot be directly applied to LBP given its discrete nature.

In this work, we consider an asymptotic analysis as the dimension of the discrete model, $N$, converges to infinity. Starting with a product distribution, we prove that the asymptotic efficiency of LBP in discrete spaces is $2R\Phi(-\frac{1}{2}\lambda_1 R^{\frac{3}{2}}/N)$ with an asymptotic acceptance rate of $2\Phi(-\frac{1}{2}\lambda_1 R^{\frac{3}{2}}/N)$, where the scale $R$ represents the number of sites to update per M-H step. Therefore, the asymptotically optimal scale of the proposal distribution is $R = O(N^{\frac{2}{3}})$ with an asymptotically optimal acceptance rate of $0.574$, independent of the target distribution. Moreover, for RWM in a discrete space, we show that the asymptotic efficiency and acceptance rate are $2R\Phi(-\frac{1}{2}\lambda_2 R^{\frac{1}{2}})$ and $2\Phi(-\frac{1}{2}\lambda_2 R^{\frac{1}{2}})$, respectively. Hence, the asymptotically optimal scale is $O(1)$ and the asymptotically optimal acceptance rate is $0.234$ for RWM. By comparing LBP and RWM at their respective optimal scales, it can be determined that LBP is $O(N^{\frac{2}{3}})$ more efficient than RWM.

These asymptotically optimal acceptance rates are robust in the following respects. First, although the initial derivation is established w.r.t. product distributions, the result can be expanded to more general distributions. Second, the efficiency is not sensitive around the optimal acceptance rate. For example, whereas $0.574$ is the optimal acceptance rate for LBP, the algorithm retains high efficiency for acceptance rates between $0.5$ and $0.7$. Based on these observations, we propose an adaptive LBP (ALBP) algorithm that automatically tunes the update scale to suit the target distribution.

We validate these theoretical findings in a series of empirical simulations on the Bernoulli model, the Ising model, factorized hidden Markov models (FHMM) and restricted Boltzmann machines (RBM). The experimental outcomes comport with the theory. Moreover, we demonstrate that ALBP can automatically find near optimal scales for these distributions. We also use ALBP to train deep energy based models (EBMs), finding that it reduces the MCMC steps needed in contrastive divergence training (Hinton, 2002; Tieleman & Hinton, 2009), significantly improving the efficiency of the overall training procedure.

## 2 Background

**Metropolis-Hastings Algorithm** Let $\pi$ denote the target distribution. Given a current state $x^{(n)}$, a M-H sampler draws a candidate state $y$ from a proposal distribution $q(x^{(n)}, y)$. Then, with probability $\min\left\{1, \frac{\pi(y)q(y, x^{(n)})}{\pi(x^{(n)})q(x^{(n)}, y)}\right\}$ the proposed state is accepted and $x^{(n+1)} = y$; otherwise, $x^{(n+1)} = x^{(n)}$. In this way, the detailed balance condition is satisfied and the M-H sampler generates a Markov chain $x_0, x_1, ...$ that has $\pi$ as its stationary distribution.

**Locally Balanced Proposal**. The locally balanced proposal (LBP) is a special case of the pointwise informed proposal (PIP), which is a class of M-H algorithms for discrete spaces (Zanella, 2020) using the proposal distribution $Q_g(x, y) \propto g(\pi(y)/\pi(x))$ such that $g$ is a scalar weight function. Zanella (2020) shows that the family of locally balancing functions $\mathcal{G} = \{g : \mathbb{R}_+ \to \mathbb{R}_+, g(t) = tg(\frac{1}{t}), \forall t > 0\}$ (e.g. $g(t) = \sqrt{t}$ or $\frac{t}{t+1}$) is asymptotically optimal for PIP. Hence, PIP with a locally balanced function for its weight function is referred to as LBP. Despite having good proposal quality, PIP requires the weight $g(\pi(z)/\pi(x))$ to be calculated for all candidate states $z$ in the neighborhood

of $x$, which results in its high computational cost. Grathwohl et al. (2021) propose to estimate the probability change by leveraging the gradient, improving the scalability of LBP.

**Locally Balanced Proposal with Auxiliary Path**. Sun et al. (2021) generalize LBP by introducing an auxiliary path sampler, which allows multiple sites to be updated per M-H step. In particular, Sun et al. (2021) sequentially selects the update indices without replacement, and uses these indices as auxiliary variables to keep the proposal distribution tractable while preserving the detailed balance condition. Although this can achieve significant improvements in empirical performance, Sun et al. (2021) manually tune the update size per M-H step, and leave the optimal scale problem open.

# 3 Main Result

## 3.1 Problem Statement

We establish asymptotic limit theorems for two M-H algorithms in discrete spaces: the *locally balanced proposal* (LBP) and *random walk Metropolis* (RWM). Following previous work (Gelman et al., 1997; Roberts & Rosenthal, 1998; Beskos et al., 2013; Vogrinc et al., 2022), we conduct our analysis on a product probability measure $\pi$. In particular, for a state space $\mathcal{X} = \{0, 1\}^N$, we consider a factored target distribution

$$\pi^{(N)}(x) = \prod_{i=1}^{N} \pi_i(x_i) = \prod_{i=1}^{N} p_i^{x_i}(1 - p_i)^{1-x_i} \tag{1}$$

where each site is assumed to have a sufficiently large probability for being both 0 and 1; that is, for a fixed $\epsilon \in (0, \frac{1}{4})$, we assume the target distribution belongs to:

$$\mathcal{P}_\epsilon := \{\pi^{(N)} : \epsilon < p_j \wedge (1 - p_j) < \frac{1}{2} - \epsilon, \forall j = 1, ..., N, N \geq 1\} \tag{2}$$

where we denote $a \wedge b = \min\{a, b\}$. To measure the efficiency of the sampler, an ergodic estimate varies with the objective function considered. Alternatively, we use a natural progress estimate: the expected jump distance (EJD). Denote $P_\theta$ as the transition kernel, $d(x, y)$ as the Hamming distance between $x$ and $y$. For a M-H sampler parameterized by $\theta$, its expected jump distance $\rho(\theta)$ and corresponding expected acceptance rate $a(\theta)$ are

$$\rho(\theta) = \sum_{X,Y \in \mathcal{X}} \pi(X) P_\theta(X, Y) d(X, Y), \quad a(\theta) = \sum_{X,Y \in \mathcal{X}} \pi(X) P_\theta(X, Y) 1_{\{X \neq Y\}} \tag{3}$$

In continuous space, the limit of sampling process is a diffusion process, whose efficiency is determined by the expected squared jump distance (ESJD) (Roberts & Rosenthal, 2001). In discrete space, the limit of the sampling process is a jump process, whose velocity is characterized by the EJD. Hence, EJD is the correct metric to measure the efficiency in discrete space; see more details in Appendix B.1.

## 3.2 Locally Balanced Proposal

We consider the M-H sampler LBP-$R$, where $R$ refers to flipping $R$ indices in each M-H step. Given a current state $x$, LBP-R calculates the weight $w_j$ for flipping index $j$ as in PIP. Since we are considering a binary target distribution of the form (1), we have

$$w_j(x) = w_j(x_j) = g\left(\frac{\pi_j(1 - x_j)}{\pi_j(x_j)}\right) \tag{4}$$

where $g$ is a locally balanced function. Following Sun et al. (2021), LBP-R select indices $u_r$ with probability $\mathbb{P}(u_r = j) \propto w_j$ sequentially for $r = 1, ..., R$, **without** replacement. The new state $y$ is obtained by flipping indices $u_{1:R}$ of $x$. If we consider $u$ as an auxiliary variable, the accept rate $A(x, y, u)$ in the M-H acceptance test can be written as

$$A(x, y, u) = 1 \wedge \frac{\pi(y) \prod_{r=1}^{R} \frac{w_{u_r}(y)}{W(y,u) + \sum_{i=1}^{r} w_{u_i}(y)}}{\pi(x) \prod_{r=1}^{R} \frac{w_{u_r}(x)}{W(x,u) + \sum_{i=r}^{R} w_{u_i}(x)}}, \quad \text{where } W(x, u) = \sum_{i=1}^{N} w_i - \sum_{r=1}^{R} w_{u_r} \tag{5}$$

From theorem 1 in Sun et al. (2021), the auxiliary sampler LBP-R satisfies detailed balance. A M-H step of LBP-R is summarized in Algorithm 1.

---

**Algorithm 1:** A M-H step of LBP-R and ALBP

1   Given current state $x^{(n)}$, current $R_t$, initialize candidate set $\mathcal{C} = \{1, .., N\}$;
2   **for** $r = 1, ..., R$ *or* $r = 1, ..., rounding(R_t)$ **do**
3      Sample $u_r$ with $\mathbb{P}(u_r = j) \propto w_j(x^{(n)}) 1_{\{j \in \mathcal{C}\}}$;
4      Pop $u_r$ out of the candidate set: $\mathcal{C} \leftarrow \mathcal{C} \backslash \{u_r\}$;
5   **end**
6   Obtain $y$ by flipping indices $u_1, ..., u_R$ of $x^{(n)}$.;
7   **if** *rand(0,1)* $< A(x^{(n)}, y, u)$ **then** $x^{(n+1)} = y$ **else** $x^{(n+1)} = x^{(n)}$;
8   **if** $t < T_{warmup}$ **then** $R_{t+1} \leftarrow R_t + (A(x^{(n)}, y, u) - 0.574)$;

---

### 3.3   Optimal Scaling for Locally Balanced Proposal

We are now ready to state the first asymptotic theorem.

**Theorem 3.1.** *For arbitrary sequence of target distributions $\{\pi^{(N)}\}_{N=1}^{\infty} \subset \mathcal{P}_\epsilon$, the M-H sampler LBP-R with a locally balanced weight function g obtains the following, if $R = \lfloor l N^{\frac{2}{3}} \rfloor$,*

$$\lim_{N \to \infty} a(R) - 2\Phi\left(-\frac{1}{2}\lambda_1 l^{\frac{3}{2}}\right) = 0 \tag{6}$$

*where $\Phi$ is the c.d.f. of standard normal distribution and $\lambda_1$ only depends on $\pi^{(N)}$*

$$\lambda_1^2 = \lambda_1^2(\pi^{(N)}) = \frac{\sum_{j=1}^N p_j w_j(1)(w_j(0) - w_j(1))^2}{4(\mathbb{E}_x[\frac{1}{N}\sum_{i=1}^N w_i(x_i)])^2 \sum_{i=1}^N p_i w_i(1)} \tag{7}$$

The definition of $\lambda_1$ in (7) explains the motivation of restricting the target distributions in (2). In fact, introducing the $\epsilon$ gives upper and lower bounds of $\lambda_1$. When all $p_j$ are arbitrarily close to $\frac{1}{2}$, $(w_j(0) - w_j(1))^2$ in numerator will be zero, so is $\lambda_1$. As a result, the acceptance rate will always be 1. Else, when all $p_j$ are arbitrarily close to 0 or 1, $\mathbb{E}_x[\frac{1}{N}\sum_{i=1}^N w_i(x_i)]$ in denominator will be zero, and $\lambda_1$ will be infinity. As a result, the acceptance rate will always be 0. So, we have to make the mild assumption in (2) to assure the following asymptotic result holds. A more detailed discussion about $\epsilon$ is given in Appendix B.2.

**Corollary 3.2.** *The optimal choice of scale for $R = l N^{\frac{2}{3}}$ is obtained when the expected acceptance rate is 0.574, independent of the target distribution.*

*Proof.* When $R = l N^{\frac{2}{3}}$, denote $z = \lambda_1^{\frac{2}{3}} l$, we have:

$$\rho(R) = a(R)R = 2l N^{\frac{2}{3}}\left(\Phi\left(-\frac{1}{2}\lambda_1 l^{\frac{3}{2}}\right) + o(1)\right) = \left(\frac{N}{\lambda_1}\right)^{\frac{2}{3}} 2z\Phi\left(-\frac{1}{2}z^{\frac{3}{2}}\right) + o\left(N^{\frac{2}{3}}\right) \tag{8}$$

It means the optimal value of $z$ is independent of the target distribution $\pi^{(N)}$. As $\Phi$ is known, we can numerically solve $z = 1.081$, and the corresponding expected acceptance rate is $a = 0.574$.     $\square$

### 3.4   Proof of Theorem 3.1

Denote the current state as $x$ and a new state proposed in LBP-R as $y$. Consider the acceptance rate $A(x, y, u)$ in (5). Using the fact that, if index $j$ is not flipped then $w_j(y) = w_j(x)$, we have:

$$\frac{\pi(y) \prod_{r=1}^R w_{u_r}(y)}{\pi(x) \prod_{r=1}^R w_{u_r}(x)} = \frac{\pi(y) \prod_{i=1}^N w_i(y)}{\pi(x) \prod_{i=1}^N w_i(x)} = \prod_{i=1}^N \frac{\pi_i(y_i)/\pi_i(x_i) g(\pi_i(x_i)/\pi_i(y_i))}{g(\pi_i(y_i)/\pi_i(x_i))} = 1 \tag{9}$$

where (9) takes advantage of the property of a locally balanced function. Hence, the acceptance rate $A(x, y, u)$ can be simplified to:

$$1 \wedge \exp \left( \sum_{r=1}^{R} \log \left( \frac{1 + \sum_{i=r}^{R} w_{u_i}(x)/W(x,u)}{1 + \sum_{i=1}^{r} w_{u_i}(y)/W(y,u)} \right) \right) \tag{10}$$

From the definition in (5), we have $W(x, u) = W(y, u)$. Denote $i \wedge j = \min\{i, j\}$ and $i \vee j = \max\{i, j\}$, we have the following approximation:

**Lemma 3.3.** *Define* $W = \mathbb{E}_{x,u}[W(x, u)]$. *We have:* $\lim_{N \to 0} \sum_{r=1}^{R} \log(\frac{1 + \sum_{i=r}^{R} w_{u_i}(x)/W(x,u)}{1 + \sum_{i=1}^{r} w_{u_i}(y)/W(y,u)}) - (A + B) = 0$, *where*

$$A = \frac{1}{W} \sum_{r=1}^{R} (R - r + 1) w_{u_i}(x_{u_i}) - r w_{u_i}(y_{u_i}) \tag{11}$$

$$B = -\frac{1}{2} \frac{1}{W^2} \sum_{i,j=1}^{R} \left[ i \wedge j \, w_{u_i}(x_{u_i}) w_{u_j}(x_{u_j}) - (R - i \vee j + 1) w_{u_i}(y_{u_i}) w_{u_j}(y_{u_j}) \right] \tag{12}$$

To analyze $A$ and $B$, we reverse the order of $x$ and $u$. In particular, instead of first sampling $x \sim \pi(x)$, then sampling $u \sim p(x|u)$, we use a reversed order where we first determine the indices $u$, then the values of $x_u$, and finally the values of $x_{-u}$.

**Lemma 3.4.** *The joint distribution* $p(x, u) = \pi(x) p(u|x)$ *can be decomposed in the following form:*

$$p(x, u) = \prod_{r=1}^{R} p(u_r | u_{1:r-1}) \prod_{r=1}^{R} p(x_{u_r} | u, x_{u_{1:r-1}}) \, p(x_{-u} | u, x_u) \tag{13}$$

*Denote* $j \notin u_{1:r-1}$ *represents* $j \neq u_i$ *for* $i = 1, ..., r - 1$, *the conditional probabilities are*

$$p(u_r = j | u_{1:r-1}) = \frac{p_j w_j(1) 1_{\{j \notin u_{1:r-1}\}}}{\sum_{i=1}^{N} p_i w_i(1) 1_{\{i \notin u_{1:r-1}\}}} + O(N^{-\frac{5}{2}}) \tag{14}$$

$$p(x_j = 1 | u, x_{1:j-1}, u_r = j) = \frac{1}{2} + r \frac{w_j(0) - w_j(1)}{W} + O(N^{-\frac{2}{3}}) \tag{15}$$

With the conditional distribution in Lemma 3.4, we are able to give a concentration property of the term $B$ and show it is safe to ignore:

**Lemma 3.5.** *With a probability larger than* $1 - O(\exp(-N^{\frac{1}{2}}))$, $B = O(N^{-\frac{1}{12}})$.

For term $A$, we use martingale central limit theorem with convergence rate (Haeusler, 1988) to bound the Kolmogorov-Smirnov statistic.

**Lemma 3.6.** *When* $R = lN^{\frac{2}{3}}$, $\lambda_1$ *defined as* (7), *we have:*

$$|\mathbb{P}(\frac{A - \mu}{\sigma} \geq t) - \Phi(t)| = O(N^{-\frac{1}{12}}), \quad \mu = -\frac{1}{2} \lambda_1^2 l^3, \quad \sigma^2 = \lambda_1^2 l^3 \tag{16}$$

By (16), the expectation w.r.t. $A$ asymptotically equals to the expectation on $\mathcal{N}(\mu, \sigma^2)$. The final step to prove Theorem 3.1 is to exploit a property of the normal distribution.

**Lemma 3.7.** *If* $Z \sim \mathcal{N}(\mu, \sigma^2)$, *then we have:*

$$\mathbb{E}[1 \wedge \exp(Z)] = \Phi\left(\frac{\mu}{\sigma}\right) + \exp\left(\mu + \frac{\sigma^2}{2}\right) \Phi\left(-\sigma - \frac{\mu}{\sigma}\right) \tag{17}$$

*where* $\Phi$ *is the c.d.f. of the standard normal distribution.*

By Lemma 3.6, 3.7, we have the expectation of (10), which is the expected accept rate, equals to:

$$\mathbb{E}[a(R)] = \Phi\left(-\frac{1}{2} \lambda_1 l^{\frac{3}{2}}\right) + \exp(0) \Phi\left(-\frac{1}{2} \lambda_1 l^{\frac{3}{2}}\right) = 2\Phi\left(-\frac{1}{2} \lambda_1 l^{\frac{3}{2}}\right) \tag{18}$$

## 3.5 Optimal Scaling for Random Walk Metropolis

We denote the Random Walk Metropolis in discrete space as RWM-$R$, where $R$ refers to flipping $R$ indices in each M-H step. Under the Bernoulli distribution, a site is more likely to stay at high probability position, so if we randomly flip a site, it is more likely to decrease its probability. That is, intuitively, the acceptance rate will decrease exponentially as the scale $R$ increases. Consequently, the optimal scale for RWM-$R$ should be $O(1)$. Though this is not a rigorous proof, the constant scaling indicates that it will be hard to directly prove an asymptotic theorem for RWM-$R$. To address this difficulty, we first restrict our target distribution to a smaller class of Bernoulli distributions $\mathcal{P}_\epsilon^{(\beta)} \subset \mathcal{P}_\epsilon$, which is formally defined as follows. For a fixed $\epsilon \in (0, \frac{1}{4})$ and a fixed $\beta > 0$, define

$$\mathcal{P}_\epsilon^{(\beta)} := \left\{ \pi^{(N)} : \frac{1}{2} - \frac{1}{2N^\beta} + \frac{\epsilon}{N^\beta} < p_j \wedge (1 - p_j) < \frac{1}{2} - \frac{\epsilon}{N^\beta} \right\} \tag{19}$$

When $N$ is large, each $p_j$ will be very close to $\frac{1}{2}$. In this way, the acceptance rate will not drop too fast when $R$ is increased, and a non-constant $R$ will be permitted. This enables us to prove:

**Theorem 3.8.** *For arbitrary sequence of target distributions $\{\pi^{(N)}\}_{N=1}^\infty \subset \mathcal{P}_\epsilon^{(\beta)}$, the M-H sampler RWM-R obtains the following, if $R = lN^{2\beta}$,*

$$\lim_{N \to \infty} a(R) - 2\Phi\left( -\frac{1}{2}\lambda_2 l^{\frac{1}{2}} \right) \tag{20}$$

*where $\Phi$ is the c.d.f. of the standard normal distribution and $\lambda_2$ only depends on $\pi^{(N)}$.*

$$\lambda_2^2 = \lambda_2^2(\pi^{(N)}) = \frac{2}{N} \sum_{i=1}^N N^{2\beta}(2p_i - 1) \log \frac{p_i}{1 - p_i} \tag{21}$$

**Corollary 3.9.** *The optimal scale $R = lN^{2\beta}$ is obtained when the expected acceptance rate is 0.234, independent of the target distribution.*

The rate in Corollary 3.9 is proved for arbitrary $\beta > 0$. If we let $\beta$ decrease to 0, at $\beta = 0$ the optimal scale for RWM-$R$ is $O(1)$ while the optimal acceptance rate is 0.234. Also, we can notice that $\mathcal{P}_\epsilon^{(\beta)}$ converges to $\mathcal{P}_\epsilon$ when $\beta$ decrease to 0 and we are able to show the optimal scale of RWM in $\mathcal{P}_\epsilon$ is $O(1)$, see details in Appendix B.3. However, this limit is not mathematically rigorous, because Theorem 3.8 and Corollary 3.9 only hold asymptotically, such that a smaller $\beta$ requires a larger $N$. Hence, when $\beta$ decreases to 0, $N$ must approach infinity to satisfy the asymptotic theorem. Although there is this minor gap in the analysis, the conclusion nevertheless aligns very well with different target distributions in the experiment section.

## 4 Adaptive Algorithm

Given knowledge of the optimal acceptance rate, one can design an adaptive algorithm that automatically tunes the scale of the M-H samplers. For this purpose, we use stochastic optimization Andrieu & Thoms (2008); Robbins & Monro (1951) to adjust the scaling parameter $R_t$ to ensure that the statistic $A_t = a_t - \delta$ approaches 0, where $a_t$ is the acceptance probability for iteration $t$ and $\delta$ is the target acceptance rate (0.574 for LBP and 0.234 for RWM). According to Theorem 3.1 and Theorem 3.8, the acceptance rate is a decreasing function of the scaling $R_t$. Hence, we use the update rule:

$$R_{t+1} \leftarrow R_t + \eta_t A_t \tag{22}$$

with step size $\eta_t = 1$. We follow common practice and adapt the tunable MCMC parameters during a warmup phase before freezing them thereafter Gelman et al. (2013). The computational cost for (22) is ignorable comparing the total cost of a M-H step. The algorithm boxes for ALBP and ARWM are given in Appendix C. More advanced implementations are possible, but it is out of the focus in the paper. We observe below that this simple approach is able to maintain the sampler robustly near the optimal acceptance rate.

# 5   Related Work

Informed proposals for Metropolis-Hastings (M-H) algorithms have been extensively studied for continuous spaces (Robert & Casella, 2013). The most famous algorithms are the Metropolis-adjusted Langevin algorithm (MALA) (Rossky et al., 1978) and Hamiltonian Monte Carlo (HMC) (Neal et al., 2011). MALA, HMC, and their variants (Girolami & Calderhead, 2011; Hoffman et al., 2014; Welling & Teh, 2011; Titsias & Dellaportas, 2019; Hirt et al., 2021; Hoffman et al., 2021; Hird et al., 2020; Livingstone & Zanella, 2019) use the gradient of the target distribution to guide the proposal distribution toward high probability regions, which brings substantial improvements in sampling efficiency compared to uninformed methods, such as random walk Metropolis (RWM) (Metropolis et al., 1953).

Informed proposals have also demonstrated recent success in discrete spaces. Zanella (2020) first gives a formal definition of the pointwise informed proposal (PIP) for discrete spaces, then proves that locally balanced proposals (LBP), using a family of locally balanced functions as the weight function in PIP, are asymptotically optimal for PIP. Following this work, Power & Goldman (2019) extended the framework to Markov jump processes and introduced non-reversible heuristics to accelerate sampling. Sansone (2021) parameterize the locally balanced function and tune it by minimizing a mutual information. Grathwohl et al. (2021) give a more scalable version of LBP for differentiable target distributions by estimating the probability change through the gradient. Despite strong empirical results, the LBP method of Zanella (2020) only flips one bit per M-H step, since PIP has to restrict the proposal distribution to a small neighborhood, e.g. a 1-Hamming ball, due to its computational cost. Sun et al. (2021) generalize LBP to flip multiple bits in a single M-H step, gaining significant improvement in sampling efficiency. However, the scaling of the proposal distribution in Sun et al. (2021) was manually tuned and the optimal scaling problem was left open.

For continuous spaces, the optimal scaling problem for informed proposals has been well studied. A significant literature has already shown that the scale can be tuned with respect to the optimal acceptance rate (Roberts & Rosenthal, 2001), e.g. 0.234 for RWM (Gelman et al., 1997), 0.574 for MALA Roberts & Rosenthal (1998), 0.651 for HMC Beskos et al. (2013), and 0.574 for Barker (Vogrinc et al., 2022), by decreasing the scale so that the Markov chain converges to a diffusion process. However, such a technique is not directly applicable to LBP given its discrete nature. Roberts (1998) make an initial attempt on discrete space, however it assumes all dimensions satisfy independent, identical Bernoulli distribution. In this work, we have established for the first time the optimal scale for LBP and RWM in discrete spaces.

# 6   Experiments

The effectiveness of LBP has been extensively demonstrated in previous work, e.g. Zanella (2020); Grathwohl et al. (2021); Sun et al. (2021), in comparison to other M-H samplers for discrete spaces, such as RWM, Gibbs sampling, the Hamming Ball sampler (Titsias & Yau, 2017), and continuous relaxation based methods Zhang et al. (2012); Pakman & Paninski (2013); Nishimura et al. (2017); Han et al. (2020). Therefore, we focus on simulating LBP-$R$, with weight function $g(t) = \frac{t}{t+1}$, and RWM-$R$ to validate our theoretical findings. More experiments, including different weight functions and comparison between "with" and "without" replacement versions of LBP are given in Appendix D.

Throughout the experiment section, we will use the gradient approximation (Grathwohl et al., 2021). That is to say, we estimate the change in probability of flipping index $i$ is estimated by: $\tilde{d}x_i = \exp((1 - 2x_i)(\nabla \log \pi(x))_i)$ For the Bernoulli distribution, this is still exact and does not hinder the justification of the theoretical results. For more complex models, this approximation makes the algorithms significantly more efficient. In particular, the gradient approximation only requires two calls of the probability function and two calls of the gradient function. Consequently, LBP with gradient approximation will take about twice time per update compared to RWM. In our experiments, we observe that LBP and GWG takes $1.2 \pm 0.2$ and $1.1 \pm 0.1$ more time per update, respectively, than RWM, across all target distributions. We therefore omit reporting the detailed run time for each method.

## 6.1 Sampling from different target distributions

We consider four target distributions: the Bernoulli distribution, the Ising model, the factorial hidden Markov model (FHMM), and the restricted Boltzmann machine (RBM). For each model, we consider three configurations: C1, C2, and C3 for smooth, moderate, and sharp target distributions. To obtain performance curves, we first simulate LBP-1 and RWM-1 for an initial acceptance rate $a_{max}$. Then, we adopt $a_{max} - 0.02, ..., a_{max} - 0.02k, ...$ as a target acceptance rate. For each rate, we use the adaptive sampler to obtain an estimated scale $R$, with which we simulate 100 chains and calculate the final real acceptance rate and efficiency. In this way, we collect abundant data points to characterize the relationship between acceptance rate and efficiency to facilitate the following analyses.

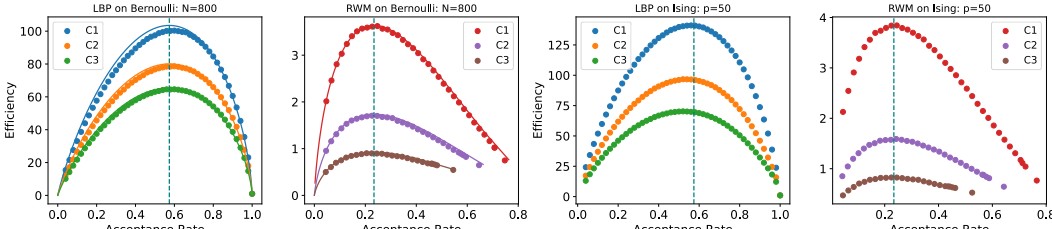

Figure 1: Efficiency Curves on Bernoulli          Figure 2: Efficiency Curves on Ising

**Bernoulli Distribution**. We validate our theoretical results on Bernoulli distribution. The probability mass function is given in (1). For each configuration, we simulate on domains with three dimensionalities: $N = 100, 800, 6400$. The scatter plot for $N = 800$ is reported in Figure 1. We also estimate $\lambda$ in (7) and (21) and plot the theoretical efficiency curve in (5) and (20). From Figure 1, we can see that the simulation results align well with the theoretically predicted curves, and the optimal efficiencies were achieved at $0.574$ for LBP and $0.234$ for RWM for all configurations.

**Ising Model**. The Ising model (Ising, 1924) is a classical model in physics defined on a $p \times p$ square lattice graph $(V_p, E_p)$ (details in Appendix D.2). For each configuration, we simulate on three sizes $p = 20, 50, 100$. We report the results for $p = 50$ in Figure 2. For LBP, the optimal efficiencies are achieved at around $0.5$, which is slightly less than $0.574$, although these values are close. Thus we can say that the asymptotically optimal acceptance rate for LBP still applies to the Ising model. For RWM, $0.234$ perfectly matches the acceptance rate where the optimal efficiencies are obtained.

**Factorial Hidden Markov Model**. The FHMM (Ghahramani & Jordan, 1995) uses latent variables $x \in \mathcal{X} = \{0, 1\}^{L \times K}$ to characterize time series data $y \in \mathbb{R}^L$ (details in Appendix D.3). Given $y$, we sample the hidden variables $x$ from the posterior $\pi(x) = p(x|y)$. For each configuration, we simulate in three sizes $L = 200, 1000, 4000$. We report the results for $L = 1000$ in Figure 3. One can observe that these results match the theoretical predictions very well.

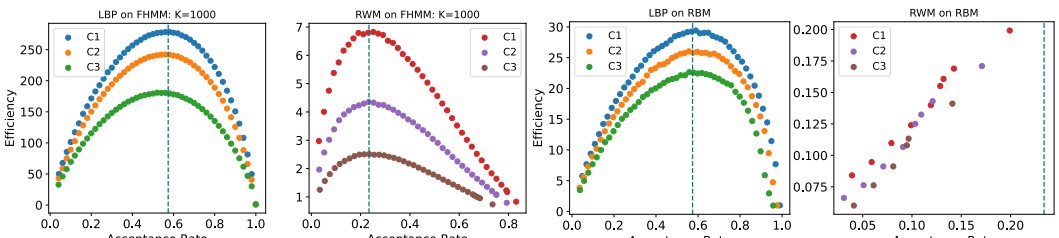

Figure 3: Efficiency Curves on FHMM          Figure 4: Efficiency Curves on RBM

**Restricted Boltzmann Machine**. A RBM (Smolensky, 1986) is a bipartite latent-variable model that defines a distribution over binary data $x \in \{0, 1\}^N$ and latent data $z \in \{0, 1\}^h$ (details in Appendix D.4). We train an RBM on the MNIST dataset using contrastive divergence (Hinton, 2002) and sample observable variables $x$. We report the results in Figure 4. For LBP, although RBM is much more complex than a product distribution, its efficiency versus acceptance rate curves still match the theoretical predictions very well. For RWM, even using $R = 1$ will result in acceptance rates less than $0.234$ for all configurations. Although we cannot check what the optimal value is, we

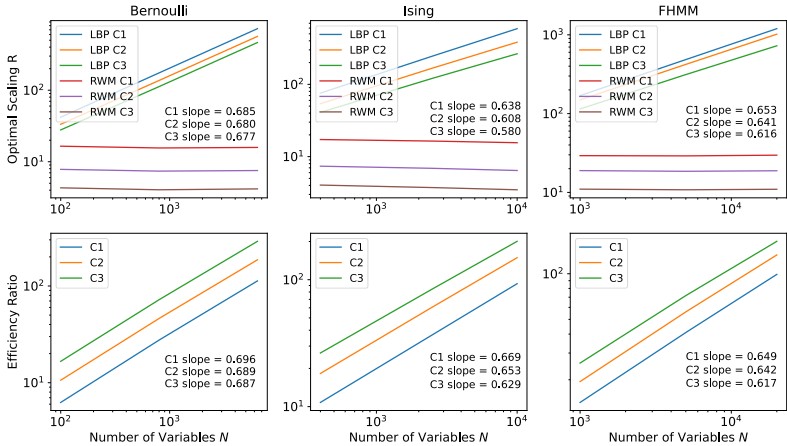

Figure 5: Optimal Scaling $R$ and Efficiency Ratio with respect to model dimension N

still observe that efficiency is an increasing function of the acceptance rate when the acceptance rate is less than $0.234$, as predicted by the theory.

**Optimal Scaling and Efficiency**. We examine how optimal scaling $R$ for LBP, RWM and their relative efficiency ratio grow w.r.t. the model dimension $N$. In figure 5, we can see that both the optimal scaling and efficiency ratio are linear in log-log plot and the slopes are close to $\frac{2}{3}$ across Bernoulli, Ising, and FHMM. The results matches the theories that the optimal scaling $R = O(N^{\frac{2}{3}})$ for LBP, $R = O(1)$ for RWM, and the relative efficiency ratio LBP over RWM is $O(N^{\frac{2}{3}})$.

Table 1: Performance of the Samplers on Various Distributions

| Size | Bernoulli | | | Ising | | | FHMM | | | RBM | | |
|---|---|---|---|---|---|---|---|---|---|---|---|---|
| Sampler | EJD | ESS | Time | EJD | ESS | Time | EJD | ESS | Time | EJD | ESS | Time |
| RWM-1 | 0.65 | 10.02 | 15.44 | 0.64 | 12.14 | 74.28 | 0.79 | 7.26 | 58.03 | 0.17 | 10.76 | 59.54 |
| ARWM | 1.70 | 18.44 | 14.90 | 1.58 | 19.60 | 77.45 | 4.32 | 13.32 | 60.02 | 0.17 | 11.13 | 61.24 |
| GRWM | 1.70 | 18.67 | 18.01 | 1.59 | 20.16 | 76.89 | 4.35 | 15.22 | 61.19 | 0.17 | 10.76 | 59.54 |
| LBP-1 | 1.00 | 13.39 | 24.36 | 1.00 | 14.11 | 111.19 | 1.00 | 6.91 | 134.42 | 0.98 | 13.38 | 116.04 |
| ALBP | 78.63 | 622.35 | 28.07 | 96.23 | 821.06 | 124.37 | 242.01 | 129.28 | 487.63 | 26.07 | 25.59 | 144.03 |
| GLBP | 78.83 | 644.43 | 25.42 | 96.68 | 809.12 | 129.28 | 242.52 | 140.43 | 508.27 | 25.86 | 25.83 | 119.38 |

## 6.2 Adaptive Sampling

We have validated the theoretical findings regarding the optimal acceptance rates on various distributions. In this section, we examine the performance of the adaptive sampler. In addition to the expected jump distance (EJD), we also report the effective sample size (ESS) [2]. We compare the adaptive sampler ALBP, ARWM with their single step version LBP-1, RWM-1, and grid search version GLBP, GRWM, where we tune the scaling $R$ by grid search. We give the sampling results on Bernoulli model, Ising model, FHMM, and RBM with medium size and configuration C2 in table 1. More results are given in Appendix D. We can see that the adaptive samplers are significantly more efficient than single step samplers, especially for LBP. Also, the adaptive samplers can robustly achieve almost the same performance comparing to using grid search to find the optimal scaling.

## 6.3 Training Deep Energy Based Models

Learning an EBM is a challenge task. Given data sampled from a true distribution $\pi$, we maximize the likelihood of the target distribution $\pi_\theta(x) \propto e^{-f_\theta(x)}$ parameterized by $\theta$. The gradient estimation requires samples from the current model, which is typically obtained via MCMC. The speed of training an EBM is determined by how fast a MCMC algorithm can obtain a good estimate of the second expectation.

---

[2]Computed using Tensorflow Probability

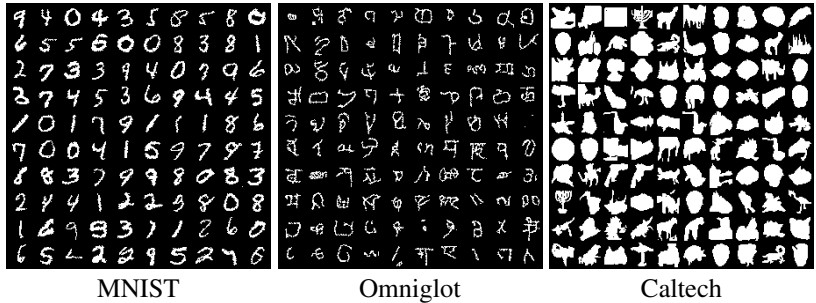

MNIST         Omniglot         Caltech

Figure 6: Samples from deep EBMs trained by ALBP$_s$ sampler.

We evaluate adaptive samplers by learning deep EBMs. Following the setting in Grathwohl et al. (2021), we train deep EBMs parameterized by Residual Networks (He et al., 2016) on small binary image datasets using PCD (Tieleman & Hinton, 2009) with a replay buffer (Du & Mordatch, 2019). We compare two single step samplers and two adaptive samplers, where LBP$_b$ uses $g(t) = \frac{t}{t+1}$ as weight function and LBP$_s$ uses $g(t) = \sqrt{t}$ as weight function. When we allow them to run enough iterations in PCD, they are able to train EBMs in same good quality. To measure the efficiency of these samplers, we compare the minimum number of M-H steps needed in PCD in table 2. We can see that adaptive samplers only need one half or even one fifth iterations compare to single step samplers. We also present long-run samples from our trained models via ALBP$_s$ in Figure 6.

Table 2: Minimum M-H Steps Needed for PCD

| Dataset | LBP$_b$-1 | ALBP$_b$ | LBP$_s$-1 | ALBP$_s$ |
|---------|-----------|----------|-----------|----------|
| Static MNIST | 90 | 20 | 40 | 15 |
| Dynamic MNIST | 100 | 20 | 40 | 15 |
| Omniglot | 100 | 60 | 30 | 5 |
| Caltech | 100 | 60 | 80 | 30 |

# 7 Discussion

In this paper, we have addressed the optimal scaling problem for the locally balanced proposal (LBP) in (Sun et al., 2021). We verified, both theoretically and empirically, that the asymptotically optimal acceptance rate for LBP is $0.574$, independent of the target distribution. Moreover, knowledge of the optimal acceptance rate allows one to adaptively tune the neighborhood size for a proposal distribution in a discrete space. We verified the theoretical findings on a diverse set of distributions, and demonstrated that adaptive LBP can improve sampling efficiency for learning deep EBMs.

We believe there is considerable room for future work that builds on these results. For theoretical investigation, the theory established under a strong assumption that the target distribution is a product distribution, despite the results applies very well to more complicated distributions. We believe the results still hold under a weaker assumption that the target distribution has no phase transition. We also believe it is possible to design a HMC style sampler for discrete spaces in the framework of Sun et al. (2021) by using LBP as a block for the auxiliary path. For empirical investigation, many real-world problems involve probability models of discrete structured data, such as syntax trees for natural language processing (Tai et al., 2015), program synthesis (Dai et al., 2020), and graphical models for molecules (Gilmer et al., 2017). Efficient discrete samplers should be able to accelerate both learning and inference with such models.

## Acknowledgement

We thank Vladimir Koltchinskii, Pengcheng Yin, Matthew D. Hoffman, and three anonymous reviewers for their helpful comments to improve the manuscript. Dale Schuurmans gratefully acknowledges the support of a Canada CIFAR AI Chair, NSERC and Amii.

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
