# A  Complete Proof

## A.1  A concentration of $W(x, u)$

**Lemma A.1.** *Define $W = \mathbb{E}_{x,u}[W(x, u)]$. We have:*

$$\mathbb{P}(|W(x, u) - W| > N^{\frac{1}{2}} t) \leq 2e^{-C_2 t^2} \tag{23}$$

*where $C_2$ is an absolute constant that only depends on the scalar $\epsilon$ in (2).*

*Proof.* Define a martingale $M_n$, $n = 0, 1, ..., N + R$. We let $M_0 = 0$. When $n \leq N$, it has independent increment

$$M_n = \sum_{i=1}^{n} w_i(x) - \mathbb{E}[w_i(x)], \quad n = 1, ..., N \tag{24}$$

For $n > N$, it is defined as

$$M_{N+r} = M_{N+r-1} - w_{u_r}(x) + \mathbb{E}[w_{u_r}(x)|M_1, ..., M_{N+r-1}] \tag{25}$$

$$= M_{N+r-1} - w_{u_r}(x) + \frac{\sum_{i \notin u_{1:r-1}} w_i^2(x)}{\sum_{i \notin u_{1:r-1}} w_i(x)} \tag{26}$$

where $i \notin u_{1:r-1}$ means $i \neq u_j$ for $j = 1, ..., r-1$. Since $p_i$ are controlled by $\epsilon$ in (2), we can find a uniform bound

$$\frac{1}{4C_1} = 2 \sup_{\epsilon < p < 1-\epsilon} g(\frac{1-p}{p}) \tag{27}$$

For $1 \leq n \leq N$, we have

$$|M_n - M_{n-1}| = |w_i(x) - \mathbb{E}[w_i(x)]| \leq 2 \max_{x,u} |w_i(x)| \leq \frac{1}{4C_1} \tag{28}$$

For $1 \leq r \leq R$, we have

$$|M_{N+r} - M_{N+r-1}| = \left| -w_{u_r}(x_{u_r}) + \frac{\sum_{i \neq u_{1:r-1}} w_i^2(x_i)}{\sum_{i \neq u_{1:r-1}} w_i(x_i)} \right| \leq \frac{1}{4C_1} \tag{29}$$

Hence, we can apply the Azuma-Hoeffding inequality:

$$\mathbb{P}(|W(x, u) - W| > tN^{\frac{1}{2}}) = \mathbb{P}(|M_n - M_0| > tN^{\frac{1}{2}}) \leq 2e^{\frac{-t^2 N}{2\frac{1}{4C_1}(N+R)}} = 2e^{-C_1 t^2}. \tag{30}$$

Thus we prove the lemma.  □

The lemma indicates with high probability, for arbitrary $\delta > 0$

$$W(x, u) - W = o(N^{\frac{1}{2}+\delta}) \tag{31}$$

One observation of the proof is that, the concentration holds for arbitrary $0 \leq R \leq N$. For example, when $R = N$, $W(x, u) \equiv W \equiv 0$, the concentration is still valid.

## A.2  Lemma 3.3

*Proof.* Using Taylor's series, we have

$$\log(1 + \sum_{i=r}^{R} w_{u_i}(x)/W(x, u)) = \frac{\sum_{i=r}^{R} w_{u_i}(x)}{W(x, u)} - \frac{1}{2}(\frac{\sum_{i=r}^{R} w_{u_i}(x)}{W(x, u)})^2 + O(\frac{R^3}{N^3}) \tag{32}$$

$$\log(1 + \sum_{i=1}^{r} w_{u_i}(y)/W(x, u)) = \frac{\sum_{i=1}^{r} w_{u_i}(y)}{W(x, u)} - \frac{1}{2}(\frac{\sum_{i=1}^{r} w_{u_i}(y)}{W(x, u)})^2 + O(\frac{R^3}{N^3}) \tag{33}$$

Using Lemma A.1 and the property $W(x, u) = W(y, u)$, with high probability, the first order term becomes to:

$$\sum_{r=1}^{R} \frac{\sum_{i=r}^{R} w_{u_i}(x)}{W(x, u)} - \frac{\sum_{i=1}^{r} w_{u_i}(y)}{W(x, u)} = \sum_{r=1}^{R} \frac{(R - r + 1)w_{u_i}(x) - r w_{u_i}(y)}{W(x, u)} \tag{34}$$

$$= \sum_{r=1}^{R} \frac{(R - r + 1)w_{u_i}(x) - r w_{u_i}(y)}{W} + O(\frac{R^2}{N^{\frac{3}{2}-\delta}}) \tag{35}$$

Similarly, with high probability, the second order term becomes to:

$$\sum_{r=1}^{R} \left(\frac{\sum_{i=r}^{R} w_{u_i}(x)}{W(x, u)}\right)^2 - \left(\frac{\sum_{i=1}^{r} w_{u_i}(y)}{W(x, u)}\right)^2 \tag{36}$$

$$= \frac{1}{W(x, u)^2} \sum_{r=r}^{R} \left( \sum_{i,j=r}^{R} w_{u_i}(x)w_{u_j}(x) - \sum_{i,j=1}^{r} w_{u_i}(y)w_{u_j}(y) \right) \tag{37}$$

$$= \frac{1}{W(x, u)^2} \sum_{i=1}^{R} \sum_{j=1}^{R} \min\{i, j\} w_{u_i}(x)w_{u_j}(x) - (R - \max\{i, j\} + 1) w_{u_i}(y)w_{u_j}(y) \tag{38}$$

$$= \frac{1}{W^2} \sum_{i=1}^{R} \sum_{j=1}^{R} \min\{i, j\} w_{u_i}(x)w_{u_j}(x) - (R - \max\{i, j\} + 1) w_{u_i}(y)w_{u_j}(y) + o(\frac{R^3}{N^{\frac{5}{2}-\delta}}) \tag{39}$$

Since $R = lN^{\frac{2}{3}}$, denote $i \wedge j = \min\{i, j\}, i \vee j = \max\{i, j\}$, with high probability, we have

$$\sum_{r=1}^{R} \log \frac{1 + \sum_{i=r}^{R} w_{u_i}(x_{u_i})/W(x, u)}{1 + \sum_{i=1}^{r} w_{u_i}(y_{u_i})/W(x, u)} \tag{40}$$

$$= \frac{1}{W} \sum_{r=1}^{R} (R - r + 1)w_{u_i}(x) - r w_{u_i}(y) + o(N^{\frac{1}{12}-\delta})$$

$$- \frac{1}{2W^2} \sum_{i=1}^{R} \sum_{j=1}^{R} i \wedge j w_{u_i}(x)w_{u_j}(x) - (R - i \vee j + 1) w_{u_i}(y)w_{u_j}(y) \tag{41}$$

Select $0 < \delta < \frac{1}{12}$, and the corresponding $t = N^{\delta}$, we have, for large enough $N$, the above equation does not hold with probability exponentially small, and the term $o(N^{\frac{1}{12}-\delta})$ can be ignored. Hence we prove the weak convergence. □

### A.3 Proof for Lemma 3.4

*Proof.* The distribution $p(u_r|u_{1:r-1})$ can be approximated using the following tricks. First, using lemma A.1, with high probability, we have:

$$\mathbb{P}(u_r = i|u_{1:r-1}) = \mathbb{E}_{x \notin u_{1:r}} \left[ \frac{\mathbb{P}(x_i = 1)w_i(1)}{W(x_{-i}, x_i = 1, u_{1:r-1})} + \frac{\mathbb{P}(x_i = 0)w_i(0)}{W(x_{-i}, x_i = 0, u_{1:r-1})} \right] \tag{42}$$

$$= \frac{p_i w_i(1) + (1 - p_i)w_i(0)}{W} + O(N^{-\frac{3}{2}}) \tag{43}$$

Derive the similar result for $\mathbb{P}(u_r = j|u_{1:r-1})$. Since we have $R = lN^{\frac{3}{2}}$, for arbitrary $1 \leq r \leq R$, we have $W$ has the same order as $N$. Using the property of locally balanced function, where $p_i w_i(1) = (1 - p_i)w_i(0)$, we have

$$\frac{\mathbb{P}(u_1 = i)}{\mathbb{P}(u_1 = j)} = \frac{p_i w_i(1)}{p_j w_j(1)} + O(N^{-\frac{5}{2}}) \tag{44}$$

Then, we use the identity:

$$1 = \sum_{i=1}^{N} \mathbb{P}(u_1 = i) \tag{45}$$

$$= \sum_{j=1}^{N} \left( \frac{p_i w_i(1)}{p_j w_j(1)} + O(N^{-\frac{5}{2}}) \right) \mathbb{P}(u_1 = j) \tag{46}$$

$$= \left( \frac{\sum_{i=1}^{N} p_i w_i(1)}{p_j w_j(1)} + O(N^{-\frac{3}{2}}) \right) \mathbb{P}(u_1 = j) \tag{47}$$

hence, we have for the first step $u_1$:

$$\mathbb{P}(u_1 = j) = \frac{p_j w_j(1)}{\sum_{i=1}^{N} p_i w_i(1)} + O(N^{-\frac{5}{2}}) \tag{48}$$

Recursively use this trick, for $1 \le r \le R = lN^{\frac{2}{3}}$ we have:

$$\mathbb{P}(u_r = j | u_{1:r-1}) = \frac{p_j w_j(1) 1_{\{j \notin u_{1:r-1}\}}}{\sum_{i=1}^{N} p_i w_i(1) 1_{\{i \notin u_{1:r-1}\}}} + O(N^{-\frac{5}{2}}) \tag{49}$$

Next, we calculate the conditional probability for $x$. To simplify the notation, we denote $\mathbb{P}(x_j = 1 | u, u_r = j, x_{u_{1:j-1}})$ to represented index $j$ is selected at step $u_r$, and not been selected in all previous steps $u_1, ..., u_{r-1}$. Also, we denote

$$W(x, u, s, t) = W(x, u) + \sum_{k=s}^{t} w_{u_k}(x) \tag{50}$$

In this way, the conditional probability for $x$ can be written as

$$\mathbb{P}(x_j = 1 | u, u_r = j, x_{u_{1:j-1}}) \tag{51}$$

$$= \mathbb{E} \left[ \frac{\pi_j(1) \prod_{l=1}^{r-1}(1 - \frac{w_j(1)}{W(x_{-j}, x_j=1, u, l, R)}) \frac{w_j(1)}{W(x_{-j}, x_j=1, u, r, R)}}{\sum_{v=0}^{1} \pi_j(v) \prod_{l=1}^{r-1}(1 - \frac{w_j(1)}{W(x_{-j}, x_j=v, u, l, R)}) \frac{w_j(1)}{W(x_{-j}, x_j=v, u, r, R)}} \bigg| u, u_r = j, x_{u_{1:j-1}} \right] \tag{52}$$

$$= \mathbb{E} \left[ \frac{\prod_{l=1}^{r-1}(1 - \frac{w_j(1)}{W(x_{-j}, x_j=1, u, l, R)}) \frac{1}{W(x_{-j}, x_j=1, u, r, R)}}{\sum_{v=0}^{1} \prod_{l=1}^{r-1}(1 - \frac{w_j(1)}{W(x_{-j}, x_j=v, u, l, R)}) \frac{1}{W(x_{-j}, x_j=v, u, r, R)}} \bigg| u, u_r = j, x_{u_{1:j-1}} \right] \tag{53}$$

Since $R = lN^{\frac{2}{3}}$, according to lemma A.1, with high probability we have:

$$\frac{w_j(1)}{W(x_{-j}, x_j = v, u, l, R)} = \frac{w_j(1)}{W + O(N^{\frac{1}{2}}) + O(R)} = \frac{w_j(1)}{W} + O(N^{-\frac{4}{3}}) \tag{54}$$

Using this approximation, we have:

$$\mathbb{P}(x_j = 1 | u, u_r = j, x_{u_{1:j-1}}) \tag{55}$$

$$= \mathbb{E} \left[ \frac{\prod_{l=1}^{r-1}(1 - \frac{w_j(1)}{W} + O(N^{-\frac{4}{3}}))(\frac{1}{W} + O(N^{-\frac{4}{3}}))}{\sum_{v=0}^{1} \prod_{l=1}^{r-1}(1 - \frac{w_j(v)}{W} + O(N^{-\frac{4}{3}}))(\frac{1}{W} + O(N^{-\frac{4}{3}}))} \bigg| u, u_r = j, x_{u_{1:j-1}} \right] \tag{56}$$

$$= \mathbb{E} \left[ \frac{\prod_{l=1}^{r-1}(1 - \frac{w_j(1)}{W} + O(N^{-\frac{4}{3}}))}{\sum_{v=0}^{1} \prod_{l=1}^{r-1}(1 - \frac{w_j(v)}{W} + O(N^{-\frac{4}{3}}))} (1 + O(N^{-\frac{2}{3}})) \bigg| u, u_r = j, x_{u_{1:j-1}} \right] \tag{57}$$

$$= \mathbb{E} \left[ \frac{1 - (r-1)\frac{w_j(1)}{W} + (r-1)O(N^{-\frac{4}{3}})}{(1 - (r-1)\frac{w_j(0)}{W}) + (1 - (r-1)\frac{w_j(1)}{W}) + (r-1)O(N^{-\frac{4}{3}})} \bigg| u, u_r = j, x_{u_{1:j-1}} \right] \tag{58}$$

$$= \mathbb{E} \left[ \frac{1 - (r-1)\frac{w_j(1)}{W}}{(1 - (r-1)\frac{w_j(0)}{W}) + (1 - (r-1)\frac{w_j(1)}{W})} + (r-1)O(N^{-\frac{4}{3}}) \bigg| u, u_r = j, x_{u_{1:j-1}} \right] \tag{59}$$

$$= \frac{1}{2} + (r-1)\frac{w_j(0) - w_j(1)}{4W} + (r-1)O(N^{-\frac{4}{3}}) \tag{60}$$

Thus we prove the lemma. $\qquad\square$

## A.4  A Property for the conditional distribution of $u$

The following result shows that marginal distribution for $u_1$ is a good approximation of the conditional distribution.

**Proposition A.2.** *For $N$ large enough, the conditional distribution for $u_r = j$ given $u_{1:r-1}$ can be approximated by the marginal distribution of $u_1$*

$$p(u_r = j | u_{1:r-1}, j \notin u_{1:r-1}) \tag{61}$$

$$= \mathbb{E}_{u_{1:r-1}} \left[ \frac{p_j w_j(1)}{\sum_{i \notin u_{1:r-1}} p_i w_i(1)} \right] + O(N^{-\frac{5}{2}}) \tag{62}$$

$$= \mathbb{E}_{u_{1:r-1}} \left[ \frac{p_j w_j(1)}{\sum_{i=1}^{N} p_i w_i(1)} + \frac{p_j w_j(1) \sum_{i=1}^{N} p_i w_i(1)(1 - 1_{\{i \notin u_{1:r-1}\}})}{(\sum_{i \notin u_{1:r-1}} p_i w_i(1))(\sum_{i=1}^{N} p_i w_i(1))} \right] + O(N^{-\frac{5}{2}}) \tag{63}$$

$$= p(u_1 = j) + O(\frac{r}{N^2}) \tag{64}$$

## A.5  Proof for Lemma 3.5

*Proof.* We first calculate its expectation using the conditional distribution derived in lemma 3.4. To simplify the notation, we denote $\delta_w(i) = w_{u_i}(0) - w_{u_i}(1)$ for $i = 1, ..., R$ and

$$S(i, j, k, l) = i \wedge j w_{u_i}(k) w_{u_j}(l) - (R - i \vee j + 1) w_{u_i}(1 - k) w_{u_j}(1 - l) \tag{65}$$

$$P(i, k) = \frac{1}{2} - (-1)^k (i - 1) \frac{\delta_w(i)}{4W} + (i - 1) O(N^{-\frac{4}{3}}) \tag{66}$$

for $i, j = 1, ..., R$, and $k, l = 0, 1$. Then we have

$$- \frac{1}{2W^2} \sum_{i=1}^{R} \sum_{j=1}^{R} [i \wedge j w_{u_i}(x_{u_i}) w_{u_j}(x_{u_j}) - (R - i \vee j + 1) w_{u_i}(y_{u_i}) w_{u_j}(y_{u_j}) | u] \tag{67}$$

$$= - \frac{1}{2W^2} \sum_{i,j=1}^{R} \sum_{k=0}^{1} \sum_{l=0}^{1} S(i, j, k, l) P(i, k) P(j, l) \tag{68}$$

$$= - \frac{1}{2W^2} \sum_{i,j=1}^{R} (R - (i + j) + 1)(w_{u_i}(0) + w_{u_i}(1))(w_{u_j}(0) + w_{u_j}(1)) + O(\frac{R^2}{N}) \tag{69}$$

$$= - \frac{1}{2W^2} \sum_{i,j=1}^{R} (R - (i + j) + 1)(w_{u_i}(0) + w_{u_i}(1))(w_{u_j}(0) + w_{u_j}(1)) + O(\frac{R^4}{N^3}) \tag{70}$$

The remaining expectation is with respect to $u$. From proposition A.2, we know that the conditional expectation of $u_i$ can be estimated via the marginal distribution of $u_1$. In fact, when $R = lN^{\frac{2}{3}}$, we have:

$$\mathbb{E}[w_{u_r}(0) + w_{u_r}(1) | u_{1:r-1}] \tag{71}$$

$$= \mathbb{E}[\sum_{j=1}^{N} (w_j(1) + w_j(0)) (\frac{p_j w_j(1)}{\sum_{i=1}^{N} p_i w_i(1)} + O(\frac{R}{N^2})) | u_{1:r-1}] \tag{72}$$

$$= \mathbb{E}[w_{u_1}(0) + w_{u_1}(1)] + O(N^{-\frac{4}{3}}) \tag{73}$$

and similarly, we have:

$$\mathbb{E}[(w_{u_r}(0) + w_{u_r}(1))^2 | u_{1:r-1}] = \mathbb{E}[(w_{u_1}(0) + w_{u_1}(1))^2] + O(N^{-\frac{4}{3}}) \tag{74}$$

Using these properties, we have

$$\mathbb{E}[\sum_{i,j=1}^{R}(R-(i+j)+1)(w_{u_i}(0)+w_{u_i}(1))(w_{u_j}(0)+w_{u_j}(1))] \tag{75}$$

$$=\mathbb{E}[\mathbb{E}[\cdots\mathbb{E}[2\sum_{i=1}^{R}\sum_{j>i}^{R}(R-(i+j)+1)(w_{u_i}(0)+w_{u_i}(1))(w_{u_j}(0)+w_{u_j}(1))$$

$$+\sum_{r=1}^{R}(R-2r+1)(w_{u_r}(0)+w_{u_r}(1))^2|u_{1:R-1}]\cdots|u_1]] \tag{76}$$

$$=\mathbb{E}[2\sum_{i=1}^{R}\sum_{j>i}^{R}(R-(i+j)+1)(w_{u_1}(0)+w_{u_1}(1))(w_{u_1}(0)+w_{u_1}(1))]+O(N^{\frac{2}{3}})$$

$$+\mathbb{E}[\sum_{r=1}^{R}(R-2r+1)(w_{u_1}(0)+w_{u_1}(1))^2]+O(1) \tag{77}$$

$$=(w_{u_1}(0)+w_{u_1}(1))^2\sum_{i,j=1}^{N}(R-(i+j)+1)+O(N^{\frac{2}{3}}) \tag{78}$$

$$=O(N^{\frac{2}{3}}) \tag{79}$$

Hence, we prove that

$$\mathbb{E}[-\frac{1}{2W^2}\sum_{i=1}^{R}\sum_{j=1}^{R}[i\wedge jw_{u_i}(x_{u_i})w_{u_j}(x_{u_j})-(R-i\vee j+1)w_{u_i}(y_{u_i})w_{u_j}(y_{u_j})]=O(N^{-\frac{4}{3}}) \tag{80}$$

The expectation of the $B$ (12) is small. To show it is save to ignore, we will prove the concentration property. Consider a function of $x$ and $u$:

$$F(x,u)=-\frac{1}{2}\frac{1}{W^2}\sum_{i=1}^{R}\sum_{j=1}^{R}[i\wedge jw_{u_i}(x_{u_i})w_{u_j}(x_{u_j})-(R-i\vee j+1)w_{u_i}(y_{u_i})w_{u_j}(y_{u_j}) \tag{81}$$

where $y$ is obtained by flipping indices $u$ of $x$. For changing $x$, we have:

$$|F(x_1,...,x_j,...,x_N,u_1,...,u_R)-F(x_1,...,x_j',...,x_N,u_1,...,u_R)|\leq c_j \tag{82}$$

where $c_j=0$ if $j\notin u$ or $c_j=O(\frac{R^2}{N^2})$ if there exists $r$ and $u_r=j$. For chaning $u$, we have

$$|F(x_1,...,x_N,u_1,...,u_i,...u_R)-F(x_1,...,x_N,u_1,...,u_i',...,u_R)|\leq d_i \tag{83}$$

where $d_i=O(\frac{R^2}{N^2})$ for $i=1,...,R$. By McDiarmid's inequality, we have:

$$\mathbb{P}(|F(x,u)-\mathbb{E}[F(x,u)]\geq t\frac{R^{\frac{5}{2}}}{N^{\frac{7}{4}}})\leq 2\exp(-\frac{2t^2R^5/N^{\frac{7}{2}}}{\sum_{j=1}^{N}c_j^2+\sum_{i=1}^{R}d_i^2})\lesssim\exp(-2t^2N^{\frac{1}{2}}) \tag{84}$$

Hence, $F(x,u)$ will concentrate to its expectation at scale $O(R^{\frac{5}{2}}/N^{\frac{7}{4}})$. Since $R=lN^{\frac{2}{3}}$, with probability larger than $1-O(\exp(-N^{\frac{1}{2}}))$, $B=O(N^{-\frac{1}{12}})$. $\qquad\square$

## A.6 Lemma 3.6

*Proof.* To show that $A$ weakly converges to a normal distribution, we use martingale central limit theorem. Define a martingale $M_n$, for $n=0,1,...,2R$. When $n\leq R$, we let the process $M_n=0$ and the filter $F_n$ as the $\sigma$-algebra determined by $u_1,...,u_n$. For $R+1\leq R+n\leq 2R$, define

$$M_{R+n}=M_{R+n-1}+\frac{1}{W}\Big((R-r+1)w_{u_n}(x_n)-rw_{u_n}(1-x_{u_n})$$

$$-\mathbb{E}[(R-r+1)w_{u_n}(x_n)-rw_{u_n}(1-x_{u_n})]\Big) \tag{85}$$

We first estimate the mean of the increment using the conditional probability derived in lemma . If $n \leq R$, the mean is obviously 0, else

$$\mathbb{E}[\frac{(R-r+1)w_{u_r}(x_{u_r}) - rw_{u_r}(y_{u_r})}{W}|u_r = j] \tag{86}$$

$$=\frac{(R-r+1)w_j(1) - rw_j(0)}{W}(\frac{1}{2} + r\frac{w_j(0) - w_j(1)}{W} + O(\frac{R}{N^{\frac{3}{2}}} + \frac{R^2}{N^2})) \tag{87}$$

$$+\frac{(R-r+1)w_j(0) - rw_j(1)}{W}(\frac{1}{2} - r\frac{w_j(0) - w_j(1)}{W} + O(\frac{R}{N^{\frac{3}{2}}} + \frac{R^2}{N^2})) \tag{88}$$

$$=\frac{1}{2}\frac{R - 2r + 1}{W}(w_j(1) + w_j(0)) - \frac{r(R+1)}{4W^2}(w_j(0) - w_j(1))^2 + O(\frac{R^2}{N^{\frac{5}{2}}} + \frac{R^3}{N^3}) \tag{89}$$

Then we estimate the variance of $M_n - M_{n-1}$. We start with estimating the 2nd moment.

$$\mathbb{E}[(\frac{(R-r+1)w_{u_r}(x_{u_r}) - rw_{u_r}(y_{u_r})}{W})^2|u_r = j] \tag{90}$$

$$=(\frac{(R-r+1)w_j(1) - rw_j(0)}{W})^2(\frac{1}{2} + r\frac{w_j(0) - w_j(1)}{W}) + O(\frac{R}{N^{\frac{3}{2}}} + \frac{R^2}{N^2})) \tag{91}$$

$$+\frac{((R-r+1)w_j(0) - rw_j(1)}{W})^2(\frac{1}{2} - r\frac{w_j(0) - w_j(1)}{W}) + O(\frac{R}{N^{\frac{3}{2}}} + \frac{R^2}{N^2})) \tag{92}$$

$$=\frac{1}{2}((R-r+1)^2 + r^2)\frac{w_j^2(0) + w_j^2(1)}{W^2} - 2r(R-r+1)\frac{w_j(0)w_j(1)}{W^2} + O(\frac{R^3}{N^{\frac{7}{2}}} + \frac{R^4}{N^4}) \tag{93}$$

Then, we are able to calculate the variance:

$$\mathrm{var}[\frac{(R-r+1)w_{u_r}(x_{u_r}) - rw_{u_r}(y_{u_r})}{W}|u_r = j] \tag{94}$$

$$=\mathbb{E}[(\frac{(R-r+1)w_{u_r}(x_{u_r}) - rw_{u_r}(y_{u_r})}{W})^2|u_r = j]$$

$$- \mathbb{E}^2[\frac{(R-r+1)w_{u_r}(x_{u_r}) - rw_{u_r}(y_{u_r})}{W}|u_r = j] \tag{95}$$

$$=\frac{(R+1)^2}{4}\frac{w_j^2(0) + w_j^2(1)}{W^2} - \frac{(R+1)^2}{2}\frac{w_j(0)w_j(1)}{W^2} + O(\frac{R^2}{N^{\frac{5}{2}}} + \frac{R^3}{N^3}) \tag{96}$$

$$=\frac{(R+1)^2}{4W^2}(w_j(0) - w_j(1))^2 + O(\frac{R^2}{N^{\frac{5}{2}}} + \frac{R^3}{N^3}) \tag{97}$$

We calculate the value of its mean $\mu$ and variance $\sigma^2$.

$$\mu = \mathbb{E}[\sum_{r=1}^{R}\frac{(R-r+1)w_{u_r}(x_{u_r}) - rw_{u_r}(y_{u_r})}{W}|u] \tag{98}$$

$$=\sum_{r=1}^{R}\frac{1}{2}\frac{R - 2r + 1}{W}(w_{u_r}(1) + w_{u_r}(0)) - \frac{r(R+1)}{4W^2}(w_{u_r}(0) - w_{u_r}(1))^2 \tag{99}$$

$$\sigma^2 = \sum_{r=1}^{R}\mathrm{var}[\frac{(R-r+1)w_{u_r}(x_{u_r}) - rw_{u_r}(y_{u_r})}{W}|u] \tag{100}$$

$$=\sum_{r=1}^{R}\frac{(R+1)^2}{4W^2}(w_{u_r}(0) - w_{u_r}(1))^2 \tag{101}$$

Define $\mu_1 = \mathbb{E}[w_{u_1}(1) + w_{u_1}(0)]$. For the first part in $\mu$, using proposition A.2, we have

$$\mathbb{E}[\sum_{r=1}^{R} \frac{R - 2r + 1}{W}(w_{u_r}(1) + w_{u_r}(0))] \tag{102}$$

$$= \mathbb{E}[\sum_{r=1}^{R} \frac{R - 2r + 1}{W}\mu_1 + O(N^{-\frac{5}{3}})] \tag{103}$$

$$= O(N^{-\frac{2}{3}}) \tag{104}$$

Define $\sigma_1^2 = \mathbb{E}[(w_{u_1}(0) - w_{u_1}(1))^2]$, From lemma 3.4, we have

$$\mathbb{E}[(w_{u_r}(0) - w_{u_r}(1))^2] = \sigma_1^2 + O(N^{-\frac{4}{3}}), \quad \forall r = 1, ..., R \tag{105}$$

for the second term in $\mu$, we have

$$\sum_{r=1}^{R} -\frac{r(R+1)}{4W^2}(w_{u_r}(0) - w_{u_r}(1))^2 = -\frac{R(R+1)^2}{8W^2}\sigma_1^2 + O(N^{-\frac{4}{3}}) \tag{106}$$

for the variance $\sigma^2$, we have:

$$\sum_{r=1}^{R} \frac{(R+1)^2}{4W^2}(w_{u_r}(0) - w_{u_r}(1))^2 = \frac{R(R+1)^2}{4W^2}\sigma_1^2 + O(N^{-\frac{4}{3}}) \tag{107}$$

Finally, we will decouple $R$ with $W$. Specifically:

$$\frac{1}{W^2} = \frac{1}{\mathbb{E}^2[\sum_{k \notin u} w_k(x_k)]} = \frac{1}{\mathbb{E}^2[\sum_{k=1}^{N} w_k(x_k)]} + O(N^{-\frac{8}{3}}) \tag{108}$$

Combine everything together, we have

$$\mu = -\frac{R(R+1)^2}{8\mathbb{E}^2[\sum_{k=1}^{N} w_k(x_k)]}\sigma_1^2 + O(N^{-\frac{2}{3}}) \tag{109}$$

$$\sigma^2 = \frac{R(R+1)^2}{4\mathbb{E}^2[\sum_{k=1}^{N} w_k(x_k)]}\sigma_1^2 + O(N^{-\frac{4}{3}}) \tag{110}$$

Since $R = lN^{\frac{2}{3}}$, we have the sum of the conditional variance is $O(1)$ and the reminder is $o(1)$. For a martingale, we need to check one more step. We know $|M_n - M_{n-1}| = 0$ for $n \le R$. For $n + R > R$, we have:

$$|M_{R+n} - M_{R+n-1}| = \frac{1}{W}((R - r + 1)w_{u_r}(x) - rw_{u_r}(y) - \mathbb{E}[(R - r + 1)w_{u_r}(x) - rw_{u_r}(y)]) \tag{111}$$

$$= O(\frac{R}{N}) = O(N^{-\frac{1}{3}}) \tag{112}$$

is uniformly bounded by a constant independent of $N$ and $R$. We denote

$$\lambda_1^2 = \frac{\sum_{j=1}^{N} p_j w_j(1)(w_j(0) - w_j(1))^2}{4\mathbb{E}^2[\frac{1}{N}\sum_{k=1}^{N} w_k(x_k)]\sum_{i=1}^{N} p_i w_i(1)} \tag{113}$$

Then we can rewrite:

$$\mu = -\frac{1}{2}\lambda_1^2 l^3 \tag{114}$$

$$\sigma^2 = \lambda_1^2 l^3 \tag{115}$$

By martingale central limit theorem, we have that:

$$\frac{A - \mu}{\sigma} \xrightarrow{\text{dist.}} \mathcal{N}(0, 1) \tag{116}$$

Furthermore, we use the convergence rate in [Haeusler](1988), we have:

$$L_{R,2\delta} \equiv \sum_{r=1}^{2R} E\left(|M_r - M_{r-1}|^{2+2\delta}\right) = O(\frac{R^{3+2\delta}}{N^{2+2\delta}}) \tag{117}$$

$$M_{R,2\delta} \equiv \mathbb{E}[|\sum_{r=1}^{2R} \mathbb{E}[(M_r - M_{r-1})^2|F_{r-1}] - 1|^{1+\delta}] = O(\frac{R^{4+4\delta}}{N^{4+4\delta}}) \tag{118}$$

Then we have the probability

$$|\mathbb{P}(\frac{A-\mu}{\sigma} \le t) - \Phi(t)| \le D_R \tag{119}$$

where

$$D_R \le C_\delta(L_{R,2\delta} + M_{R,2\delta})^{\frac{1}{3+2\delta}} = O(R/N^{\frac{2+2\delta}{3+2\delta}}), \quad \forall \delta > 0 \tag{120}$$

where $C_\delta$ is an absolute constant that only depends on $\delta$. We select $\delta = \frac{1}{2}$, we have:

$$|\mathbb{P}(\frac{A-\mu}{\sigma} \le t) - \Phi(t)| \le O(R/N^{\frac{3}{4}}) \tag{121}$$

Since we consider $R = lN^{\frac{2}{3}}$, we prove the lemma. $\qquad\square$

## A.7 Proof of Lemma 3.7

*Proof.* Assume $Z \sim \mathcal{N}(\mu, \sigma^2)$, then we have:

$$\mathbb{E}\min\{1, e^Z\} = \int_{-\infty}^0 e^z \frac{1}{\sqrt{2\pi}\sigma} e^{-\frac{(z-\mu)^2}{2\sigma^2}} dz + \int_0^\infty \frac{1}{\sqrt{2\pi}\sigma} e^{-\frac{(z-\mu)^2}{2\sigma^2}} dz \tag{122}$$

$$= \int_{-\infty}^0 \frac{1}{\sqrt{2\pi}\sigma} e^{-\frac{z^2-2\mu z+\mu^2-2\sigma^2 z}{2\sigma^2}} dz + \int_{-\mu}^\infty \frac{1}{\sqrt{2\pi}\sigma} e^{-\frac{z^2}{2\sigma^2}} dz \tag{123}$$

$$= \exp(\mu + \frac{\sigma^2}{2}) \int_{-\infty}^0 \frac{1}{\sqrt{2\pi}\sigma} e^{-\frac{(z-(\mu+\sigma^2))^2}{2\sigma^2}} dz + \int_{-\mu}^\infty \frac{1}{\sqrt{2\pi}\sigma} e^{-\frac{z^2}{2\sigma^2}} dz \tag{124}$$

$$= \exp(\mu + \frac{\sigma^2}{2}) \int_{-\infty}^{-\mu-\sigma^2} \frac{1}{\sqrt{2\pi}\sigma} e^{-\frac{z^2}{2\sigma^2}} dz + \int_{-\infty}^\mu \frac{1}{\sqrt{2\pi}\sigma} e^{-\frac{z^2}{2\sigma^2}} dz \tag{125}$$

$$= \exp(\mu + \frac{\sigma^2}{2}) \Phi(-\frac{\mu}{\sigma} - \sigma) + \Phi(\frac{\mu}{\sigma}) \tag{126}$$

Specially, when $\mu = -\frac{1}{2}\sigma^2$, we have:

$$\mathbb{E}\min\{1, e^Z\} = 2\Phi(-\frac{1}{2}\sigma) \tag{127}$$

$\qquad\square$

## A.8 Proof for Theorem 3.8

*Proof.* In RWM-R, the proposal distribution is uniform, hence we only need to consider the probability ratio in the acceptance rate. Given current state $x$ and the picked indices $u$, the proposed state $y$ is

obtained by flipping indices $u$ of $x$. The acceptance rate is:

$$A(x, y, u) = 1 \wedge \frac{\pi(y)}{\pi(x)} \tag{128}$$

$$= 1 \wedge \prod_{r=1}^{R} \frac{\pi_{u_r}(y)}{\pi_{u_r}(x)} \tag{129}$$

$$= 1 \wedge \prod_{r=1}^{R} \frac{p_{u_r}^{y_{u_r}} (1 - p_{u_r})^{1 - y_{u_r}}}{p_{u_r}^{x_{u_r}} (1 - p_{u_r})^{1 - x_{u_r}}} \tag{130}$$

$$= 1 \wedge \prod_{r=1}^{R} p_{u_r}^{1 - 2x_{u_r}} (1 - p_{u_r})^{2x_{u_r} - 1} \tag{131}$$

$$= 1 \wedge \exp\left(\sum_{r=1}^{R} (1 - 2x_{u_r}) \log \frac{p_{u_r}}{1 - p_{u_r}}\right) \tag{132}$$

Define the martingale $M_n$, $n = 1, ..., 2R$. For $r = 1, ..., R$, we have $M_r = 0$ and the filtration $F_r$ is determined by the $\sigma$-algebra of $u_1, ..., u_R$. For $R + 1 \le R + n \le 2R$, we have:

$$M_{R+n} = M_{R+n-1} + (1 - 2x_{u_n}) \log \frac{p_{u_n}}{1 - p_{u_n}} - \mathbb{E}[(1 - 2x_{u_n}) \log \frac{p_{u_n}}{1 - p_{u_n}}] \tag{133}$$

Hence, for $n \le R$, the increment is 0. For $n + R > R$, denote the mean of the increment is :

$$\mathbb{E}[(1 - 2x_{u_n}) \log \frac{p_{u_n}}{1 - p_{u_n}}] = (1 - 2p_{u_n}) \log \frac{p_{u_n}}{1 - p_{u_n}} \tag{134}$$

the variance of the increment is:

$$\mathbb{E}[(M_{R+j} - M_{R+j-1})^2 | u, x_{1:j-1}] \tag{135}$$

$$= \mathbb{E}[((1 - 2x_{u_n}) \log \frac{p_{u_n}}{1 - p_{u_n}} - \mathbb{E}[(1 - 2x_{u_n}) \log \frac{p_{u_n}}{1 - p_{u_n}}])^2] \tag{136}$$

$$= \mathbb{E}[((1 - 2x_{u_n}) \log \frac{p_{u_n}}{1 - p_{u_n}})^2] - \mathbb{E}^2[(1 - 2x_{u_n}) \log \frac{p_{u_n}}{1 - p_{u_n}}] \tag{137}$$

$$= (\log \frac{p_{u_n}}{1 - p_{u_n}})^2 - (1 - 2p_{u_n})^2 (\log \frac{p_{u_n}}{1 - p_{u_n}})^2 \tag{138}$$

$$= 4p_j (1 - p_{u_n})(\log \frac{p_{u_n}}{1 - p_{u_n}})^2 \tag{139}$$

When $N$ is large, we have $p_{u_n} - \frac{1}{2} = O(N^{-\beta})$, hence

$$4p_{u_n}(1 - p_{u_n})(\log \frac{p_{u_n}}{1 - p_{u_n}})^2 \tag{140}$$

$$= 4p_{u_n}(1 - p_{u_n}) \log(1 + \frac{2p_{u_n} - 1}{p_{u_n}}) \log(\frac{p_{u_n}}{1 - p_{u_n}}) \tag{141}$$

$$= 4(\frac{1}{2} + O(N^{-\beta}))(1 - p_{u_n})(\frac{2p_{u_n} - 1}{1 - p_{u_n}} + O(N^{-2\beta})) \log(\frac{p_{u_n}}{1 - p_{u_n}}) \tag{142}$$

$$= 2(2p_{u_n} - 1) \log(\frac{p_{u_n}}{1 - p_{u_n}})(1 + O(N^{-\beta})) \tag{143}$$

is negative twice of the corresponding mean. Since the indices $u$ are uniformly picked, the conditional distribution of $u_r$ is:

$$\mathbb{P}(u_r = j | u_{1:r-1}) = \frac{1_{\{j \notin u_{1:r-1}\}}}{\sum_{i=1}^{N} 1_{\{i \notin u_{1:r-1}\}}} = \frac{1}{N} + O(\frac{R}{N^2}) \tag{144}$$

Hence, we have the mean is

$$\mu = \mathbb{E}[\sum_{r=1}^{R}(1 - 2x_{u_n})\log\frac{p_{u_n}}{1 - p_{u_n}}] \tag{145}$$

$$= \mathbb{E}[R(1 - 2x_{u_1})\log\frac{p_{u_1}}{1 - p_{u_1}} + O(\frac{R^2}{N^2})] \tag{146}$$

$$= \frac{R}{N^{2\beta}}\frac{1}{N}\sum_{i=1}^{N}N^{2\beta}(1 - 2p_i)\log\frac{p_i}{1 - p_i} + O(\frac{R^2}{N^2}) \tag{147}$$

Similarly, we have the variance is:

$$\sigma^2 = \mathbb{E}[\sum_{r=1}^{R}2(2x_{u_n} - 1)\log\frac{p_{u_n}}{1 - p_{u_n}}] = \frac{R}{N^{2\beta}}\frac{2}{N}\sum_{i=1}^{N}N^{2\beta}(2p_i - 1)\log\frac{p_i}{1 - p_i} + O(\frac{R^2}{N^2}) \tag{148}$$

When $R = O(N^{2\beta})$, the variance is at a constant order. For a martingale, we also need to check the increments are uniformly bounded. When $n \leq R$, the increment is always 0. When $R + 1 \leq R + n \leq 2R$, we have:

$$|M_{R+n} - M_{R+n-1}| = |(1 - 2x_{u_n})\log\frac{p_{u_n}}{1 - p_{u_n}} - \mathbb{E}[(1 - 2x_{u_n})\log\frac{p_{u_n}}{1 - p_{u_n}}]| \leq C(\epsilon) \tag{149}$$

where $C(\epsilon)$ is a constant only determined by $\epsilon$. Hence, by martingale central limit theorem, we have the distribution of $M_{2R}$ converges to a normal distribution. Denote

$$\lambda_2^2 = \frac{2}{N}\sum_{i=1}^{N}N^{2\beta}(2p_i - 1)\log\frac{p_i}{1 - p_i} \tag{150}$$

Then we can rewrite:

$$\mu = -\frac{1}{2}\lambda_2^2\frac{R}{N^{2\beta}} \tag{151}$$

$$\sigma^2 = \lambda_2^2\frac{R}{N^{2\beta}} \tag{152}$$

Denote $Z = \sum_{r=1}^{R}(1 - 2x_{u_r})\log\frac{p_{u_r}}{1 - p_{u_r}}$. By martingale central limit theorem, we have

$$\frac{Z - \mu}{\sigma} \longrightarrow_{\text{dist.}} \mathcal{N}(0, 1) \tag{153}$$

Furthermore, using the convergence rate in [Haeusler (1988)](#), we have:

$$L_{R,2\delta} \equiv \sum_{r=1}^{2R}E\left(|M_r - M_{r-1}|^{2+2\delta}\right) = O(\frac{R}{N^{(4+4\delta)\beta}}) \tag{154}$$

$$M_{R,2\delta} \equiv \mathbb{E}[|\sum_{r=1}^{2R}\mathbb{E}[(M_r - M_{r-1})^2|F_{r-1}] - 1|^{1+\delta}] = O(\frac{R^{2+2\delta}}{N^{2+2\delta}}) \tag{155}$$

Then we have the probability

$$|\mathbb{P}(\frac{A - \mu}{\sigma} \leq t) - \Phi(t)| \leq D_R \tag{156}$$

where

$$D_R \leq C_\delta(L_{R,2\delta} + M_{R,2\delta})^{\frac{1}{3+2\delta}} = O(R^{\frac{1}{3+2\delta}}/N^{\frac{4+4\delta}{3+2\delta}}), \quad \forall\delta > 0 \tag{157}$$

where $C_\delta$ is an absolute constant that only depends on $\delta$. We select $\delta = \frac{1}{2}$, we have:

$$|\mathbb{P}(\frac{A - \mu}{\sigma} \leq t) - \Phi(t)| \leq O(R^{\frac{1}{4}}/N^{\frac{5}{4}}) \tag{158}$$

Hence, the expectation w.r.t. $\sum_{r=1}^{R}(1 - 2x_{u_r})\log\frac{p_{u_r}}{1 - p_{u_r}}$ converges to the expectation w.r.t.

$$\mathcal{N}(-\frac{1}{2}\lambda_2^2\frac{R}{N^{2\beta}}, \lambda_2^2\frac{R}{N^{2\beta}}) \tag{159}$$

Using lemma 3.7, we have the acceptance rate converges to:

$$a(R) = 2\Phi(-\frac{1}{2}\lambda_2 \frac{R^{\frac{1}{2}}}{N^\beta}) \tag{160}$$

In RWM-R, the distance between the current state $x$ and the proposed state $y$ is always $d(x, y) = R$, hence we have:

$$\rho(R) = Ra(R) = 2R\Phi(-\frac{1}{2}\lambda_2 \frac{R^{\frac{1}{2}}}{N^\beta}) \tag{161}$$

When $R = \omega(N^{2\beta})$, we can give a concentration property. Since the selection of $u_r$ is a martingale, we can apply Azuma-Hoeffding inequality:

$$\mathbb{P}(|M_{2R} - \mu| > t\lambda_2 R^{\frac{3}{4}}/N^{\frac{3}{2}\beta}) \lesssim 2\exp(-\frac{2t^2 R^{\frac{3}{2}}/N^{3\beta}}{RN^{-2\beta}}) = 2\exp(-2t^2 R^{\frac{1}{2}}/N^\beta) \tag{162}$$

Hence, When $N$ is sufficiently large, with probability larger than $1 - O(\exp(-2t^2 R^{\frac{1}{2}}/N^\beta))$, we have:

$$\sum_{r=1}^{R}(1 - 2x_{u_r})\log\frac{p_{u_r}}{1 - p_{u_r}} = -\frac{1}{2}\lambda_2^2 \frac{R}{N^{2\beta}} + O(\frac{tR^{\frac{3}{4}}}{N^{\frac{3}{2}\beta}}) = -\frac{C}{2}\lambda_2^2 \frac{R}{N^{2\beta}} \tag{163}$$

For $C > 0$ independent with $N, R$. $\qquad\square$

## A.9  Proof for Corollary 3.9

*Proof.* When $R = O(N^{2\beta})$, denote $z = R\lambda_2^2/N^{2\beta}$

$$\rho(R) = 2R\Phi(-\frac{1}{2}\lambda_2 \frac{R^{\frac{1}{2}}}{N^\beta}) \tag{164}$$

$$= 2(N^{2\beta}/R)(R\lambda_2^2/N^{2\beta})\Phi(-\frac{1}{2}((R\lambda_2^2/N^{2\beta})^{\frac{1}{2}}) \tag{165}$$

$$= 2(N^{2\beta}/R)z\Phi(-\frac{1}{2}z^{\frac{1}{2}}) \tag{166}$$

which means the optimal value of $z$ is independent of the target distribution. As $\Phi$ is known, we can numerically solve $z = 5.673$. Hence the corresponding expected acceptance rate $a = 0.234$, independent with the target distribution, and the efficiency is $\Theta(N^{2\beta})$. When $R = \omega(N^{2\beta})$, with probability $1 - O(\exp(-2R/N^\beta))$, the acceptance rate decrease exponentially fast, rendering $o(1)$ jump distance. For the remaining probability $O(\exp(-2R/N^\beta))$, assuming all proposals are accepted, the efficiency is still bounded by:

$$R\exp(-2R/N^\beta) = o(1) \tag{167}$$

Hence, optimal efficiency is achieved when $R = O(N^{2\beta})$. $\qquad\square$

# B   Discussion

## B.1   Expected Jump Distance as the Metric to Tune the Scale

In this section, we want to convince the reader that the expected jump distance (EJD) is the correct metric to evaluate the efficiency for samplers in discrete space. To simplify the derivation, we consider the distribution

$$\pi^{(N)}(x) = \prod_{i=1}^{N} \pi_i(x_i) = \prod_{i=1}^{N} p^{x_i}(1-p)^{1-x_i} \tag{168}$$

We can notice that, compared to the target distributions considered in the main text (1), we assume the target distribution is identical in each dimension.

Let the LBP chain, with $R = lN^{\frac{2}{3}}$, being denoted as $\{x(1), x(2), ...\}$. Since all dimensions are identical, we only need to focus on the first dimension. Denote $w_1 = g(\frac{\pi_1(x_1=0)}{\pi_1(x_1=1)})$ and $w_0 = g(\frac{\pi_1(x_1=1)}{\pi_1(x_1=0)})$. From Lemma. A.1, we can see that:

$$\lim_{N\to\infty} \frac{\mathbb{P}(u, \exists u_j = 1 | x_1 = 0)}{\mathbb{P}(u, \exists u_j = 1 | x_1 = 1)} = \frac{w_0}{w_1} \tag{169}$$

That's to say, the probability ratio of $x_1 = 0$ and $x_1 = 1$ being flipped equals to their weight ratio. Then we compare the acceptance rate in M-H test. From the proof of the main theorem 3.1, we know the acceptance rate is determined by the term $A$ defined in (11)

$$A = \frac{1}{W} \sum_{r=1}^{R} (R - r + 1)w_{u_i}(x_{u_i}) - rw_{u_i}(y_{u_i}) \tag{170}$$

We can see that, when the first dimension is flipped in proposal, the difference of $A$ is $O(N^{-\frac{1}{3}})$ for $x_1 = 0$ and $x_1 = 1$. As a result, we have:

$$\lim_{N\to\infty} \frac{\mathbb{P}(\text{accept } | u, \exists u_j = 1, x_1 = 0)}{\mathbb{P}(\text{accept } | u, \exists u_j = 1, x_1 = 1)} = 1 \tag{171}$$

Now, we consider the one-dimensional process $Z_t^N = x_1(\lfloor tN^{\frac{1}{3}} \rfloor)$. The identical assumption implies that, the frequency for a site, for example the first dimension, being selected is $lN^{-\frac{1}{3}}$. We can easily see that when $N$ is large enough, $Z_t^N$ converges to a jump process $Z_t$, whose generator we denote.

$$Q = \begin{bmatrix} -Q_{01} & Q_{01} \\ Q_{10} & -Q_{10} \end{bmatrix} \tag{172}$$

From the derivation above, we know that

$$\frac{Q_{01}}{Q_{10}} = \lim_{N\to\infty} \frac{\sum_u \mathbb{E}_{x_{2:N}}[\mathbb{P}(u, \exists u_j = 1 | x_1 = 0)\mathbb{P}(\text{accept } | u, \exists u_j = 1, x_1 = 0)]}{\sum_u \mathbb{E}_{x_{2:N}}[\mathbb{P}(u, \exists u_j = 1 | x_1 = 1)\mathbb{P}(\text{accept } | u, \exists u_j = 1, x_1 = 1)]} = \frac{w_0}{w_1} \tag{173}$$

Since the sketch of proof above shows that the ratio is independent with the parameter $l$, we have the following important decomposition

$$Q = \lambda(l)Q(p) \tag{174}$$

where $Q(p)$ is a matrix only depends on $p$ and the locally balanced function $g$ selected, and $\lambda(l)$ is a scalar only depends on the parameter $l$.

Since $Q(p)$ only depends on the target distribution, for any test functions $f(\cdot)$, the inverse auto-correlation of the jump process is proportional to $\lambda(l)$. When we tune $l$, the coefficient $\lambda(l) = l \cdot 2\Phi(-\lambda_1 l^{\frac{3}{2}})$ is the multiplication of the proposal frequency and the acceptance rate. The value $\lambda_1$ is defined in (7). As a jump process, we don't have to analytically compute the value of $\lambda(l)$, as $\lambda(l)$ is proportional to the expected jump distance (EJD). So, we can tune $l$ by maximizing the EJD, without having to know the formulation of the target distribution.

Remark 1: The jump process in discrete space is different from the diffusion process in continuous space. For diffusion process, its velocity is characterized by the ESJD. But for jump process, its

| $\epsilon$ | 0.1 | 0.05 | 0.025 | 0.0125 |
|---|---|---|---|---|
| $N$ | 10 | 40 | 160 | 640 |

Table 3: When $p = 0.5 - \epsilon$

| $\epsilon$ | 0.01 | 0.005 | 0.0025 | 0.00125 |
|---|---|---|---|---|
| $N$ | 50 | 100 | 200 | 400 |

Table 4: When $p = \epsilon$

velocity is characterized by the EJD. That's why Langevin algorithms tunes the step size via ESJD (Roberts & Rosenthal, 1998), but our LBP tunes the path length via EJD.

Remark 2: To simplify the derivation, we assume that the target distributions have identical marginals. For target distributions with non-identical marginal distribution, different dimensions $i = 1, ..., N$ can have different velocity $\lambda_i(l)$. But the sampling process will still converge to jump process, and we shall still use EJD to measure the efficiency.

## B.2 The Choice of $\epsilon$ and the Optimal Acceptance Rate

The convergence of (6) does not depend on the value of $\epsilon$ in (2). Based on the proof above, we can know (6) converges at the rate $O(N^{-\frac{1}{12}})$. But the convergence of the optimal acceptance rate depends on the $\epsilon$. We can first consider two extreme cases for intuition. When all $p_i$ are close to $\frac{1}{2}$, $\lambda_1$ in (7) will be close to 0 and the optimal acceptance rate will be close to 1; when all $p_i$ are close to 0 or 1, $\lambda_1$ in (7) will be close to $\infty$ and the optimal acceptance rate will be close to 0. Hence, the main purpose to use fixed $\epsilon$ is to give upper and lower bounds for $\lambda_1$ in (7), such that the optimal acceptance rate can converge to 0.574 as in Corollary 3.2.

Next, we discuss how does the model dimension $N$ in (1) needed in terms of $\epsilon$ to make sure the optimal convergence to 0.574. When all $p_i$ have the extreme value determined by $\epsilon$, using locally balanced function $g(t) = \sqrt{t}$, we can consider the following two situations:

- All $|p_i - 0.5| = \epsilon \to 0$. Then we have:

$$\lambda_1^2 = \frac{\sum_{i=1}^{N} \sqrt{\epsilon(1-\epsilon)}(\sqrt{\frac{\epsilon}{1-\epsilon}} - \sqrt{\frac{1-\epsilon}{\epsilon}})^2}{4\epsilon(1-\epsilon)\sum_{i=1}^{N}\sqrt{\epsilon(1-\epsilon)}} \approx \frac{\sum_{i=1}^{N} 0.5 \cdot 4\epsilon^2}{4 \cdot \sum_{i=1}^{N} 0.5} = \epsilon^2 \quad (175)$$

  When the expected acceptance is 0.574, we need $\lambda_1 l^{\frac{3}{2}} = O(\epsilon l^{\frac{3}{2}})$ equals to a constant, which means $l$ has the same order as $\epsilon^{-\frac{2}{3}}$. Since we have $R = lN^{\frac{2}{3}} \leq N$, we need $\epsilon^{-\frac{2}{3}} = O(N^{\frac{1}{3}})$. As a result, we requires $\epsilon^{-1} = O(N^{\frac{1}{2}})$, which basically means we need $N \geq \epsilon^{-2}$ to have the optimal acceptance rate converges to 0.574.

- All $0.5 - |p_i - 0.5| = \epsilon \to 0$. We have:

$$\lambda_1^2 = \frac{\sum_{j=1}^{N} \epsilon\sqrt{\frac{1-\epsilon}{\epsilon}}(\sqrt{\frac{1-\epsilon}{\epsilon}} - \sqrt{\frac{\epsilon}{1-\epsilon}})^2}{4(\sqrt{\epsilon(1-\epsilon)})^2\sum_{i=1}^{N}\sqrt{\epsilon(1-\epsilon)}} \approx \frac{\sum_{j=1}^{N}\epsilon^{\frac{1}{2}}\epsilon^{-1}}{4\epsilon\sum_{j=1}^{N}\epsilon^{-\frac{1}{2}}} = \frac{1}{4}\epsilon^{-2} \quad (176)$$

  When the expected acceptance is 0.574, we need $\lambda_1 l^{\frac{3}{2}} = O(\epsilon^{-1}l^{\frac{3}{2}})$ equals to a constant, which means $l$ has the same order as $\epsilon^{\frac{2}{3}}$. Since we have $R = lN^{\frac{2}{3}} \geq 1$, we have $l^{-1} = O(N^{\frac{2}{3}})$. As a result, we requires $\epsilon^{-1} = O(N^{-1})$, which basically means we need $N \geq \epsilon^{-1}$ to have the optimal acceptance rate converges to 0.574.

So, both situations show that we need $N$ increase with $\epsilon$ to make sure the optimal acceptance rate converges. In the main text, we assume $\epsilon$ is a constant and it guarantees Corollary 3.2 holds.

We conduct extra numerical simulations to verify our results. To simplify the experiments, we assume all dimensions have the same configurations: $p_i = p$. We report the size of $N$ needed to guarantee that the optimal acceptance rate is 0.574 in Table 3 and Table 4.

## B.3 Optimal Scale of RWM

When we assume the target distribution belongs to (2), the derivation of the optimal acceptance rate 0.234 is no longer valid. But we can still show the optimal scale is $R = O(1)$ by proving the acceptance rate decreasing exponentially fast.

In particular, assume we use $R = lN^\beta$ in RWM. Then the acceptance rate can be written as:

$$A = \min\{1, A' = \frac{\pi(y)}{\pi(x)} = \frac{\prod_{j=1}^{R} \pi_{u_j}(y)}{\prod_{j=1}^{R} \pi_{u_j}(x)}\} \tag{177}$$

Consider the martingale $M_j, j = 0, 1, ..., R$, such that $M_0 = 0$ and

$$M_j - M_{j-1} = \log \frac{\pi_{u_j}(y)}{\pi_{u_j}(x)} - \mathbb{E}[\log \frac{\pi_{u_j}(y)}{\pi_{u_j}(x)}|u_{1:j-1}] = (1 - 2x_{u_j}) \log \frac{p_{u_j}}{1 - p_{u_j}} \tag{178}$$

By assumption in (2), we know that

$$\mathbb{E}[\log \frac{\pi_{u_j}(y)}{\pi_{u_j}(x)}|u_{1:j-1}] = \mathbb{E}[(1 - 2x_{u_j}) \log \frac{p_{u_j}}{1 - p_{u_j}}|u_{1:j-1}] = (1 - 2p_{u_j}) \log \frac{p_{u_j}}{1 - p_{u_j}} \tag{179}$$

$$\leq 2\epsilon \log \frac{1 - 2\epsilon}{1 + 2\epsilon} < 0 \tag{180}$$

And we have

$$|M_j - M_{j-1}| \leq 2 \left|(1 - 2\epsilon) \log \frac{\epsilon}{1 - \epsilon}\right| := K \tag{181}$$

By Azuma-Hoeffding lemma, we have

$$\mathbb{P}(|\sum_{j=1}^{R} \log \frac{\pi_{u_j}(y)}{\pi_{u_j}(x)} - \mathbb{E}[\log \frac{\pi_{u_j}(y)}{\pi_{u_j}(x)}]| \geq R^{\frac{3}{4}}t) \leq 2e^{\frac{-Rfrac12t^2}{K^2}} \tag{182}$$

For $\beta > 0$, $R$ increases to infinity when $N \to \infty$. In this case, $\log A'$ concentrates to a value $T \leq R \cdot 2\epsilon \log \frac{1-2\epsilon}{1+2\epsilon}$ and $A'$ decreases exponentially fast. Hence, the optimal scaling of RWM is $O(1)$.

## C Adaptive Algorithm

We give the algorithm box for ALBP:

---
**Algorithm 2:** Adaptive Locally Balanced Proposal

---
1: Initialize current state $x^{(1)}$.
2: Initialize scale $R_1 = 1$.
3: **for** t=1, ..., T **do**
4:     Initialize candidate set $\mathcal{C} = \{1, .., N\}$.
5:     $R \leftarrow$ probabilistic rounding of $R_t$
6:     **for** r=1, ..., R **do**
7:         Sample $u_r$ with $\mathbb{P}(u_r = j) \propto w_j(x^{(t)})1_{\{j \in \mathcal{C}\}}$.
8:         $\mathcal{C} \leftarrow \mathcal{C}\backslash\{u_r\}$.
9:     **end for**
10:    Obtain $y$ by flipping indices $u_1, ..., u_R$ of $x^{(t)}$.
11:    Compute acceptance rate $A = A(x^{(t)}, y, u)$.
12:    **if** rand(0,1) $< A$ **then**
13:       $x^{(t+1)} = y$
14:    **else**
15:       $x^{(t+1)} = x^{(t)}$
16:    **end if**
17:    **if** $t < T_{\text{warmup}}$ **then**
18:       $R_{t+1} \leftarrow R_t + (A - 0.574)$.
19:    **end if**
20: **end for**

---

We give the algorithm box for ARWM:

---
**Algorithm 3:** Adaptive Random Walk Metropolis

---
1: Initialize current state $x^{(1)}$.
2: Initialize scale $R_1 = 1$.
3: **for** t=1, ..., T **do**
4:     Initialize candidate set $\mathcal{C} = \{1, .., N\}$.
5:     $R \leftarrow$ probabilistic rounding of $R_t$
6:     Uniformly sample $u_1, ..., u_R$.
7:     Obtain $y$ by flipping indices $u_1, ..., u_R$ of $x^{(t)}$.
8:     Compute acceptance rate $A = A(x^{(t)}, y, u)$.
9:     **if** rand(0,1) $< A$ **then**
10:      $x^{(t+1)} = y$
11:    **else**
12:       $x^{(t+1)} = x^{(t)}$
13:    **end if**
14:    **if** $t < T_{\text{warmup}}$ **then**
15:       $R_{t+1} \leftarrow R_t + (A - 0.234)$.
16:    **end if**
17: **end for**

---

# D  Experiment Details

We consider five samplers:

- RWM: random walk Metropolis
- GWG($\sqrt{t}$): LBP with replacement, same as algorithm 1 except for skipping line 5, weight function $g(t) = \sqrt{t}$
- LBP($\sqrt{t}$): LBP given in algorithm 1, weight function $g(t) = \sqrt{t}$
- GWG($\frac{t}{t+1}$): LBP with replacement, same as algorithm 1 except for skipping line 5, weight function $g(t) = \frac{t}{t+1}$
- LBP($\frac{t}{t+1}$): LBP given in algorithm 1, weight function $g(t) = \frac{t}{t+1}$

For each sampler, we first start simulating with $R = 1$ to get an initial acceptance rate $a_{\max}$. Then we adopt $a_{\max} - 0.02, a_{\max} - 0.04, ..., a_{\max} - 0.02k$ as the target acceptance rate, until $a_{\max} - 0.02k < 0.03$. For each target acceptance rate $a_{\text{target}}$, we use our adaptive sampler to get an estimated scaling $R_{\text{target}}$. Then we simulate 100 chains with scaling $R_{\text{target}}$ to get the expected acceptance rate, expected jump distance, effective sample size $(a, d, e)$.

To measure the performance of the adaptive sampler, we compare three versions for each sampler above. In particular, for sampler X we have

- X-1, represents fixed scaling $R = 1$ version of the sampler.
- AX, represents the adaptive version of the sampler, whose target acceptance rate is selected as $0.234$ for RWM, and $0.574$ for else.
- GX, represents the grid search version of the sampler, where we always use the best results among all simulations for different target acceptance rates we mentioned above.

## D.1  Simulation on Bernoulli Model

The density function for Bernoulli distribution is:

$$\pi^{(N)}(x) = \prod_{i=1}^{N} \pi_i(x_i) = \prod_{i=1}^{N} p_i^{x_i} (1 - p_i)^{1-x_i} \tag{183}$$

We consider three configurations

- C1: $p_i$ is independently, uniformly sampled from $[0.25, 0.75]$.
- C2: $p_i$ is independently, uniformly sampled from $[0.15, 0.85]$.
- C3: $p_i$ is independently, uniformly sampled from $[0.05, 0.95]$.

For each configuration, we simulate on three sizes:

- $N = 100$, sample Markov chain $x_{1:10000}$, use $x_{1:5000}$ for burn in, use $x_{5001:10000}$ to estimate expected acceptance rate, expected jump distance, effective sample size.
- $N = 800$, sample Markov chain $x_{1:40000}$, use $x_{1:20000}$ for burn in, use $x_{20001:40000}$ to estimate expected acceptance rate, expected jump distance, effective sample size.
- $N = 6400$, sample Markov chain $x_{1:100000}$, use $x_{1:50000}$ for burn in, use $x_{50001:100000}$ to estimate expected acceptance rate, expected jump distance, effective sample size.

We give the scatter plot of $(a, d)$ and $(a, r)$ in figure 7. And we examine the performance of our adaptive algorithm in table 5 and table 6.

## D.2  Simulation on Ising Model

Ising model is a classic model in physics defined on a $p \times p$ square lattice graph $(V_p, E_p)$. That's to say, the nodes are indexed by $\{1, ..., p\}^2$ and an edge $((i, j), (k, l))$ exists if and only if one of the following condition holds:

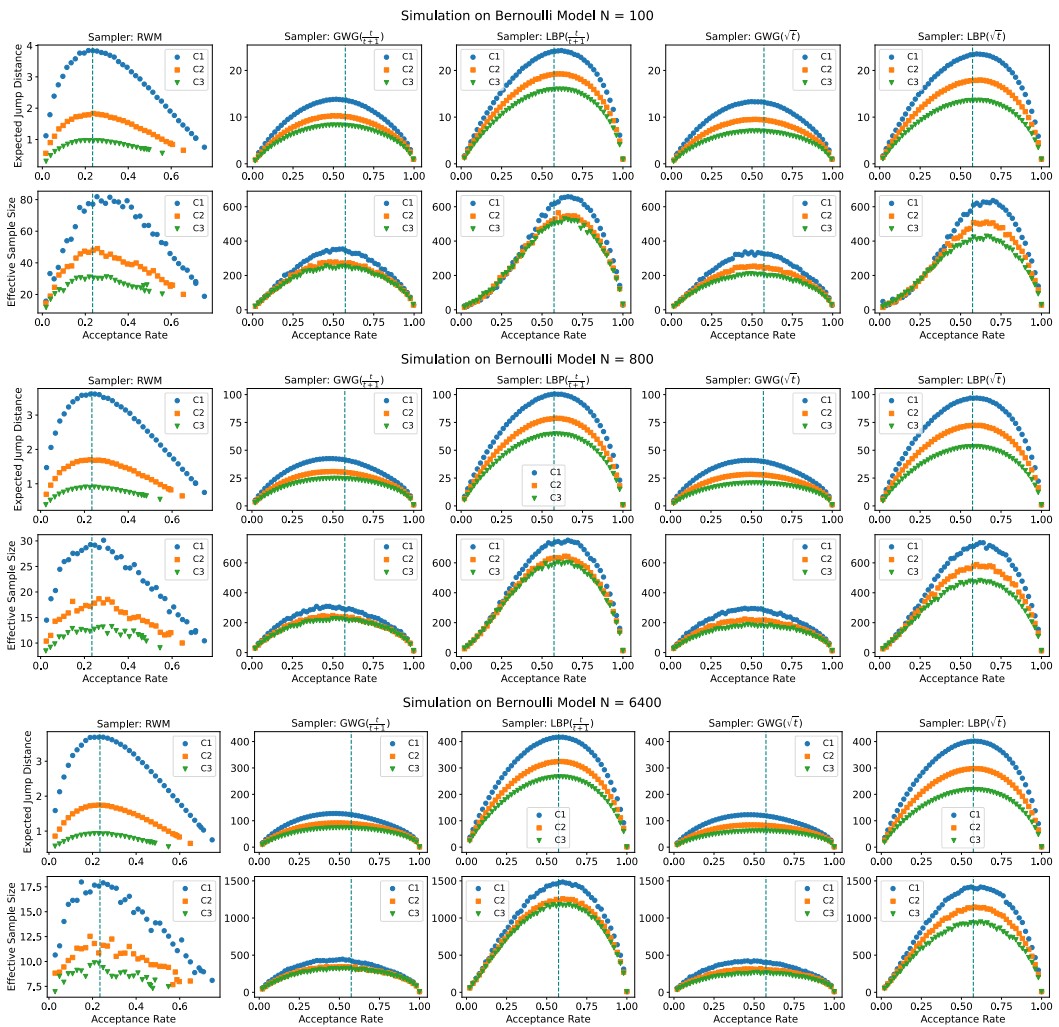

Figure 7: Simulation Results on Bernoulli Model

- $i = k,\ j = l + 1$
- $i = k + 1,\ j = l$
- $i = k,\ j = l - 1$
- $i = k - 1,\ j = l$

The state space is $\mathcal{X} = \{-1, 1\}^{V_p}$ and the target distribution is defined as:

$$\pi(x) \propto \exp\Big( \sum_{i \in V_p} \alpha_i x_i - \lambda \sum_{(i,j) \in E_p} x_i x_j \Big) \tag{184}$$

Following Zanella (2020), we consider three configurations

- C1: $\alpha_v$ is independently and uniformly sampled from $(-0.2, 0.4)$ if $(v_1 - \frac{p}{2})^2 + (v_2 - \frac{p}{2})^2 \leq \frac{p^2}{2}$, else $\alpha_v$ is independently and uniformly sampled from $(-0.4, 0.2)$; $\lambda = 0.1$.
- C2: $\alpha_v$ is independently and uniformly sampled from $(-0.3, 0.6)$ if $(v_1 - \frac{p}{2})^2 + (v_2 - \frac{p}{2})^2 \leq \frac{p^2}{2}$, else $\alpha_v$ is independently and uniformly sampled from $(-0.6, 0.3)$; $\lambda = 0.15$.
- C3: $\alpha_v$ is independently and uniformly sampled from $(-0.4, 0.8)$ if $(v_1 - \frac{p}{2})^2 + (v_2 - \frac{p}{2})^2 \leq \frac{p^2}{2}$, else $\alpha_v$ is independently and uniformly sampled from $(-0.8, 0.4)$; $\lambda = 0.2$.

For each configuration, we simulate on three sizes:

| Size | $N=100$ | | | $N=800$ | | | $N=6400$ | | |
|---|---|---|---|---|---|---|---|---|---|
| Sampler | C1 | C2 | C3 | C1 | C2 | C3 | C1 | C2 | C3 |
| RWM-1 | 0.75 | 0.66 | 0.56 | 0.75 | 0.65 | 0.55 | 0.75 | 0.65 | 0.55 |
| ARWM | 3.85 | 1.81 | 0.96 | 3.63 | 1.70 | 0.89 | 3.69 | 1.74 | 0.92 |
| GRWM | 3.84 | 1.83 | 0.96 | 3.61 | 1.70 | 0.90 | 3.69 | 1.75 | 0.93 |
| GWG($\frac{t}{t+1}$)-1 | 1.00 | 1.00 | 1.00 | 1.00 | 1.00 | 1.00 | 1.00 | 1.00 | 1.00 |
| AGWG($\frac{t}{t+1}$) | 13.60 | 10.14 | 8.26 | 41.45 | 30.23 | 24.39 | 123.08 | 89.48 | 72.56 |
| GGWG($\frac{t}{t+1}$) | 13.84 | 10.30 | 8.31 | 42.50 | 30.88 | 24.74 | 127.55 | 91.68 | 73.33 |
| LBP($\frac{t}{t+1}$)-1 | 1.00 | 1.00 | 1.00 | 1.00 | 1.00 | 1.00 | 1.00 | 1.00 | 1.00 |
| ALBP($\frac{t}{t+1}$) | 24.05 | 19.16 | 15.96 | 100.25 | 78.63 | 64.47 | 416.55 | 324.59 | 266.73 |
| GLBP($\frac{t}{t+1}$) | 24.26 | 19.26 | 15.98 | 100.49 | 78.83 | 64.69 | 416.67 | 324.67 | 266.21 |
| GWG($\sqrt{t}$)-1 | 1.00 | 1.00 | 0.99 | 1.00 | 1.00 | 1.00 | 1.00 | 1.00 | 1.00 |
| AGWG($\sqrt{t}$) | 13.14 | 9.41 | 6.92 | 39.88 | 27.81 | 20.46 | 118.44 | 82.52 | 61.27 |
| GGWG($\sqrt{t}$) | 13.31 | 9.52 | 7.04 | 40.92 | 28.34 | 20.60 | 122.74 | 84.08 | 61.52 |
| LBP($\sqrt{t}$)-1 | 1.00 | 1.00 | 1.00 | 1.00 | 1.00 | 1.00 | 1.00 | 1.00 | 1.00 |
| ALBP($\sqrt{t}$) | 23.40 | 17.88 | 13.59 | 96.96 | 72.39 | 53.28 | 401.94 | 297.95 | 218.09 |
| GLBP($\sqrt{t}$) | 23.53 | 17.95 | 13.61 | 96.93 | 72.41 | 53.36 | 401.89 | 297.72 | 218.11 |

Table 5: Expected Jump Distance on Bernoulli Model

| Size | $N=100$ | | | $N=800$ | | | $N=6400$ | | |
|---|---|---|---|---|---|---|---|---|---|
| Sampler | C1 | C2 | C3 | C1 | C2 | C3 | C1 | C2 | C3 |
| RWM-1 | 18.85 | 20.09 | 20.27 | 10.44 | 10.02 | 9.06 | 8.11 | 8.04 | 7.46 |
| ARWM | 80.86 | 47.95 | 30.49 | 28.54 | 18.44 | 12.97 | 17.97 | 11.25 | 8.66 |
| GRWM | 81.82 | 49.00 | 31.11 | 30.13 | 18.67 | 13.22 | 17.99 | 12.53 | 9.86 |
| GWG($\frac{t}{t+1}$)-1 | 28.11 | 27.89 | 31.56 | 10.80 | 12.75 | 13.61 | 7.93 | 9.17 | 9.00 |
| AGWG($\frac{t}{t+1}$) | 343.74 | 270.00 | 253.53 | 302.65 | 234.94 | 215.58 | 423.86 | 334.06 | 307.75 |
| GGWG($\frac{t}{t+1}$) | 353.97 | 278.47 | 255.11 | 309.04 | 247.38 | 227.08 | 446.20 | 343.49 | 320.93 |
| LBP($\frac{t}{t+1}$)-1 | 27.37 | 30.62 | 33.31 | 12.11 | 13.39 | 14.19 | 8.81 | 9.06 | 9.57 |
| ALBP($\frac{t}{t+1}$) | 604.07 | 528.66 | 511.27 | 754.69 | 622.35 | 594.59 | 1472.86 | 1247.31 | 1185.65 |
| GLBP($\frac{t}{t+1}$) | 658.24 | 564.55 | 529.03 | 751.22 | 644.43 | 604.47 | 1484.93 | 1259.07 | 1179.93 |
| GWG($\sqrt{t}$)-1 | 26.19 | 30.17 | 30.92 | 12.35 | 12.41 | 14.29 | 8.97 | 8.97 | 9.00 |
| AGWG($\sqrt{t}$) | 335.66 | 254.36 | 205.81 | 284.31 | 206.75 | 175.17 | 406.81 | 303.40 | 261.13 |
| GGWG($\sqrt{t}$) | 336.70 | 254.30 | 209.78 | 296.11 | 224.29 | 187.23 | 422.89 | 318.11 | 267.72 |
| LBP($\sqrt{t}$)-1 | 28.36 | 27.88 | 30.93 | 12.11 | 12.50 | 14.12 | 8.48 | 10.07 | 9.36 |
| ALBP($\sqrt{t}$) | 598.66 | 488.24 | 412.74 | 702.91 | 570.50 | 482.58 | 1411.88 | 1135.84 | 935.89 |
| GLBP($\sqrt{t}$) | 636.34 | 510.63 | 428.40 | 734.46 | 588.45 | 482.09 | 1417.15 | 1147.37 | 946.22 |

Table 6: Effective Sample Size on Bernoulli Model

- $p=20$, sample Markov chain $x_{1:10000}$, use $x_{1:5000}$ for burn in, use $x_{5001:10000}$ to estimate expected acceptance rate, expected jump distance, effective sample size.

- $p=50$, sample Markov chain $x_{1:40000}$, use $x_{1:20000}$ for burn in, use $x_{20001:40000}$ to estimate expected acceptance rate, expected jump distance, effective sample size.

- $p=100$, sample Markov chain $x_{1:100000}$, use $x_{1:50000}$ for burn in, use $x_{50001:100000}$ to estimate expected acceptance rate, expected jump distance, effective sample size.

We give the scatter plot of $(a,d)$ and $(a,r)$ in figure 8. And we examine the performance of our adaptive algorithm in table 8 and table 9.

| Size | $N = 100$ | | | $N = 800$ | | | $N = 6400$ | | |
|---|---|---|---|---|---|---|---|---|---|
| Sampler | C1 | C2 | C3 | C1 | C2 | C3 | C1 | C2 | C3 |
| RWM-1 | 6.26 | 8.85 | 5.15 | 14.45 | 15.44 | 12.48 | 39.10 | 42.70 | 41.07 |
| ARWM | 7.10 | 6.32 | 5.27 | 15.84 | 14.90 | 14.74 | 42.18 | 43.27 | 44.44 |
| GRWM | 7.33 | 6.41 | 5.90 | 13.66 | 18.01 | 14.56 | 43.64 | 44.58 | 42.24 |
| GWG($\frac{t}{t+1}$)-1 | 7.63 | 9.17 | 7.45 | 13.61 | 13.13 | 12.73 | 60.10 | 66.24 | 54.25 |
| AGWG($\frac{t}{t+1}$) | 7.49 | 9.61 | 7.48 | 14.18 | 17.95 | 15.59 | 57.39 | 72.89 | 69.93 |
| GGWG($\frac{t}{t+1}$) | 9.07 | 9.04 | 8.15 | 16.89 | 20.09 | 10.99 | 70.31 | 68.87 | 63.86 |
| LBP($\frac{t}{t+1}$)-1 | 10.60 | 12.68 | 10.35 | 23.58 | 24.36 | 19.74 | 70.11 | 94.35 | 86.53 |
| ALBP($\frac{t}{t+1}$) | 11.30 | 10.83 | 11.83 | 24.81 | 28.07 | 21.57 | 129.27 | 108.42 | 108.88 |
| GLBP($\frac{t}{t+1}$) | 10.76 | 11.57 | 10.97 | 30.71 | 25.42 | 19.33 | 92.47 | 100.16 | 100.06 |
| GWG($\sqrt{t}$)-1 | 10.80 | 11.22 | 7.19 | 17.70 | 26.87 | 19.60 | 58.95 | 59.23 | 61.19 |
| AGWG($\sqrt{t}$) | 9.57 | 13.00 | 7.23 | 18.22 | 19.46 | 20.27 | 78.60 | 67.59 | 65.13 |
| GGWG($\sqrt{t}$) | 9.44 | 7.36 | 8.64 | 18.83 | 20.72 | 20.11 | 65.64 | 50.45 | 65.91 |
| LBP($\sqrt{t}$)-1 | 13.18 | 13.82 | 11.71 | 25.62 | 28.37 | 28.94 | 86.08 | 81.86 | 90.00 |
| ALBP($\sqrt{t}$) | 12.51 | 12.21 | 10.59 | 23.16 | 22.51 | 21.02 | 102.78 | 107.33 | 103.70 |
| GLBP($\sqrt{t}$) | 14.38 | 13.48 | 10.34 | 23.97 | 30.38 | 21.48 | 120.14 | 100.18 | 117.96 |

Table 7: Running Time on Bernoulli Model

| Size | $p = 20$ | | | $p = 50$ | | | $p = 100$ | | |
|---|---|---|---|---|---|---|---|---|---|
| Sampler | C1 | C2 | C3 | C1 | C2 | C3 | C1 | C2 | C3 |
| RWM-1 | 0.77 | 0.65 | 0.54 | 0.77 | 0.64 | 0.52 | 0.76 | 0.63 | 0.51 |
| ARWM | 4.02 | 1.70 | 0.89 | 3.83 | 1.58 | 0.82 | 3.64 | 1.47 | 0.76 |
| GRWM | 4.12 | 1.69 | 0.90 | 3.84 | 1.59 | 0.83 | 3.64 | 1.47 | 0.76 |
| GWG($\frac{t}{t+1}$)-1 | 1.00 | 1.00 | 1.00 | 1.00 | 1.00 | 1.00 | 1.00 | 1.00 | 1.00 |
| AGWG($\frac{t}{t+1}$) | 27.93 | 19.35 | 14.73 | 74.33 | 50.27 | 37.68 | 150.31 | 100.21 | 74.15 |
| GGWG($\frac{t}{t+1}$) | 28.39 | 19.64 | 14.96 | 76.33 | 51.44 | 38.31 | 155.29 | 102.49 | 75.47 |
| LBP($\frac{t}{t+1}$)-1 | 1.00 | 1.00 | 1.00 | 1.00 | 1.00 | 1.00 | 1.00 | 1.00 | 1.00 |
| ALBP($\frac{t}{t+1}$) | 43.42 | 30.99 | 23.50 | 141.01 | 96.23 | 69.54 | 338.14 | 219.73 | 152.05 |
| GLBP($\frac{t}{t+1}$) | 43.45 | 31.10 | 23.62 | 141.20 | 96.68 | 70.16 | 339.11 | 221.49 | 154.52 |
| GWG($\sqrt{t}$)-1 | 1.00 | 1.00 | 1.00 | 1.00 | 1.00 | 1.00 | 1.00 | 1.00 | 1.00 |
| AGWG($\sqrt{t}$) | 26.96 | 18.09 | 13.29 | 72.04 | 47.25 | 34.30 | 145.43 | 94.41 | 68.09 |
| GGWG($\sqrt{t}$) | 27.49 | 18.32 | 13.39 | 73.94 | 48.03 | 34.54 | 149.84 | 95.79 | 68.30 |
| LBP($\sqrt{t}$)-1 | 1.00 | 1.00 | 1.00 | 1.00 | 1.00 | 1.00 | 1.00 | 1.00 | 1.00 |
| ALBP($\sqrt{t}$) | 43.85 | 31.83 | 24.32 | 146.02 | 105.43 | 79.82 | 364.76 | 261.05 | 195.65 |
| GLBP($\sqrt{t}$) | 43.73 | 31.76 | 24.33 | 146.21 | 105.38 | 79.78 | 364.84 | 260.81 | 195.67 |

Table 8: Expected Jump Distance on Ising Model

### D.3   Simulation on FHMM

FHMM uses latent variables $x \in \mathcal{X} = \{0,1\}^{L \times K}$ to characterize time series data $y \in \mathbb{R}^L$. Denote $p(x)$ as the prior for hidden variables $x$, and $p(y|x)$ for the likelihood:

$$p(x) = \prod_{l=1}^{L} p(x_{l,1}) \prod_{k=2}^{K} p(x_{l,k}|x_{l,k-1}) \tag{185}$$

$$p(y|x) = \prod_{l=1}^{L} \mathcal{N}(y_l; w^T x_l + b, \sigma^2) \tag{186}$$

Specifically, we have $p(x_{l,1}) = 0.1$, $p(x_{l,k} = x_{l,k-1}|x_{l,k-1}) = 0.8$ independently $\forall l = 1, ..., L$ and $\forall k = 2, ..., K$. And we have all entries in $W$ and $b$ are independent Gaussian random variables. We sample latent variables $x$ and sample $y \sim p(y|x)$. Then we simulate our samplers to sample the latent variables $x$ from the posterior $\pi(x) = p(x|y)$.

We consider three configurations

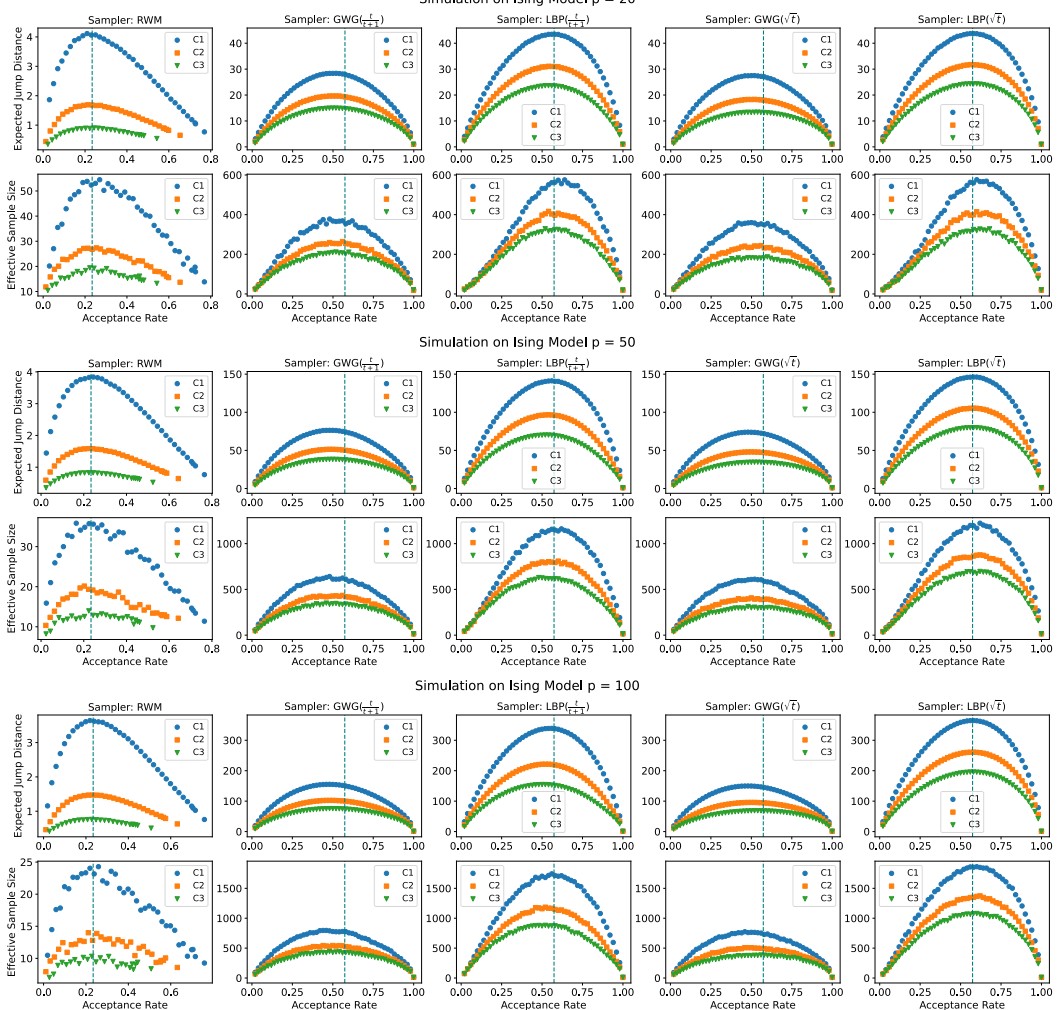

Figure 8: Simulation Results on Ising Model

- C1: $\sigma^2 = 2$

- C2: $\sigma^2 = 1$

- C3: $\sigma^2 = 0.5$

For each configuration, we simulate on three sizes:

- $L = 200, K = 5$, sample Markov chain $x_{1:10000}$, use $x_{1:5000}$ for burn in, use $x_{5001:10000}$ to estimate expected acceptance rate, expected jump distance, effective sample size.

- $L = 1000, K = 5$, sample Markov chain $x_{1:40000}$, use $x_{1:20000}$ for burn in, use $x_{20001:40000}$ to estimate expected acceptance rate, expected jump distance, effective sample size.

- $L = 4000, K = 5$, sample Markov chain $x_{1:100000}$, use $x_{1:50000}$ for burn in, use $x_{50001:100000}$ to estimate expected acceptance rate, expected jump distance, effective sample size.

We give the scatter plot of $(a, d)$ and $(a, r)$ in figure 9. And we examine the performance of our adaptive algorithm in table 11 and table 12.

| Size | $p = 20$ | | | $p = 50$ | | | $p = 100$ | | |
|---|---|---|---|---|---|---|---|---|---|
| Sampler | C1 | C2 | C3 | C1 | C2 | C3 | C1 | C2 | C3 |
| RWM-1 | 13.85 | 13.70 | 13.25 | 11.39 | 12.14 | 9.73 | 9.27 | 8.58 | 8.36 |
| ARWM | 51.66 | 27.50 | 17.39 | 35.34 | 19.60 | 12.89 | 22.99 | 13.31 | 9.47 |
| GRWM | 54.48 | 27.36 | 19.41 | 35.85 | 20.16 | 13.96 | 24.28 | 13.99 | 10.32 |
| GWG($\frac{t}{t+1}$)-1 | 19.55 | 18.06 | 20.73 | 13.53 | 13.30 | 14.54 | 10.26 | 11.44 | 11.38 |
| AGWG($\frac{t}{t+1}$) | 362.96 | 250.15 | 205.55 | 611.15 | 429.22 | 340.15 | 755.44 | 533.07 | 419.56 |
| GGWG($\frac{t}{t+1}$) | 377.87 | 264.00 | 211.38 | 641.09 | 434.17 | 349.96 | 795.99 | 543.68 | 441.19 |
| LBP($\frac{t}{t+1}$)-1 | 17.78 | 18.76 | 20.24 | 13.69 | 14.11 | 14.88 | 10.04 | 10.32 | 12.30 |
| ALBP($\frac{t}{t+1}$) | 551.81 | 394.65 | 330.29 | 1135.03 | 821.06 | 620.26 | 1733.51 | 1164.64 | 868.62 |
| GLBP($\frac{t}{t+1}$) | 575.40 | 416.56 | 328.42 | 1161.62 | 809.12 | 629.38 | 1742.19 | 1184.58 | 880.69 |
| GWG($\sqrt{t}$)-1 | 19.95 | 17.66 | 17.87 | 13.22 | 13.57 | 14.22 | 9.54 | 10.17 | 11.50 |
| AGWG($\sqrt{t}$) | 356.57 | 236.55 | 176.42 | 569.23 | 399.21 | 306.83 | 727.25 | 501.14 | 379.41 |
| GGWG($\sqrt{t}$) | 359.85 | 244.00 | 186.74 | 611.60 | 407.16 | 312.81 | 774.36 | 508.04 | 384.21 |
| LBP($\sqrt{t}$)-1 | 18.24 | 19.32 | 20.65 | 14.01 | 14.09 | 16.07 | 10.24 | 11.53 | 11.08 |
| ALBP($\sqrt{t}$) | 562.14 | 413.21 | 329.44 | 1197.76 | 867.77 | 680.61 | 1866.85 | 1359.86 | 1078.16 |
| GLBP($\sqrt{t}$) | 576.18 | 414.54 | 328.02 | 1223.24 | 877.04 | 695.96 | 1861.60 | 1374.12 | 1079.32 |

Table 9: Effective Sample Size on Ising Model

| Size | $p = 20$ | | | $p = 50$ | | | $p = 100$ | | |
|---|---|---|---|---|---|---|---|---|---|
| Sampler | C1 | C2 | C3 | C1 | C2 | C3 | C1 | C2 | C3 |
| RWM-1 | 18.78 | 19.73 | 19.64 | 71.79 | 74.28 | 75.45 | 173.58 | 142.63 | 143.42 |
| ARWM | 18.84 | 19.94 | 19.37 | 76.05 | 77.45 | 78.17 | 134.24 | 149.44 | 150.26 |
| GRWM | 19.20 | 20.08 | 19.99 | 76.09 | 76.89 | 76.90 | 134.70 | 149.48 | 150.13 |
| GWG($\frac{t}{t+1}$)-1 | 29.54 | 31.07 | 31.78 | 89.62 | 92.75 | 97.31 | 228.38 | 224.22 | 228.92 |
| AGWG($\frac{t}{t+1}$) | 31.07 | 32.54 | 40.28 | 97.31 | 99.73 | 104.38 | 271.45 | 304.41 | 273.28 |
| GGWG($\frac{t}{t+1}$) | 31.10 | 32.38 | 32.26 | 96.96 | 99.61 | 104.46 | 271.39 | 267.24 | 273.17 |
| LBP($\frac{t}{t+1}$)-1 | 36.40 | 37.61 | 46.76 | 108.31 | 111.19 | 116.87 | 260.65 | 291.16 | 269.27 |
| ALBP($\frac{t}{t+1}$) | 37.42 | 38.36 | 38.82 | 126.34 | 124.37 | 116.80 | 308.16 | 320.08 | 317.61 |
| GLBP($\frac{t}{t+1}$) | 37.91 | 39.26 | 39.12 | 124.74 | 129.28 | 125.60 | 309.07 | 310.78 | 316.89 |
| GWG($\sqrt{t}$)-1 | 29.93 | 30.59 | 31.06 | 115.42 | 120.13 | 121.43 | 216.42 | 237.03 | 240.41 |
| AGWG($\sqrt{t}$) | 30.17 | 31.68 | 31.53 | 95.37 | 98.34 | 103.86 | 261.55 | 273.21 | 280.95 |
| GGWG($\sqrt{t}$) | 30.57 | 31.46 | 31.44 | 121.69 | 125.99 | 127.66 | 259.48 | 304.54 | 280.64 |
| LBP($\sqrt{t}$)-1 | 36.99 | 37.42 | 36.98 | 106.82 | 110.86 | 117.53 | 258.04 | 275.03 | 283.44 |
| ALBP($\sqrt{t}$) | 37.62 | 38.84 | 37.58 | 125.48 | 128.87 | 121.34 | 303.28 | 306.60 | 312.44 |
| GLBP($\sqrt{t}$) | 36.87 | 38.96 | 37.87 | 125.95 | 130.17 | 174.29 | 403.14 | 305.27 | 339.78 |

Table 10: Running Time on Ising Model

## D.4 Simulation on RBM

RBM is a bipartite latent-variable model, defining a distribution over binary data $x \in \{0,1\}^N$ and latent data $z \in \{0,1\}^h$. Given parameters $W \in \mathbb{R}^{h \times N}, b \in \mathbb{R}^N, c \in \mathbb{R}^h$, the distribution of observable variables $x$ is obtained by marginalizing the latent variables $z$:

$$\pi(x) \propto \exp(b^T x) \prod_{i=1}^{h} (1 + \exp(W_i x + c_i)) \tag{187}$$

We train the RBM on MNIST dataset using contrastive divergence (Hinton, 2002) in three configurations

- C1: $h = 100$
- C2: $h = 400$
- C3: $h = 1000$

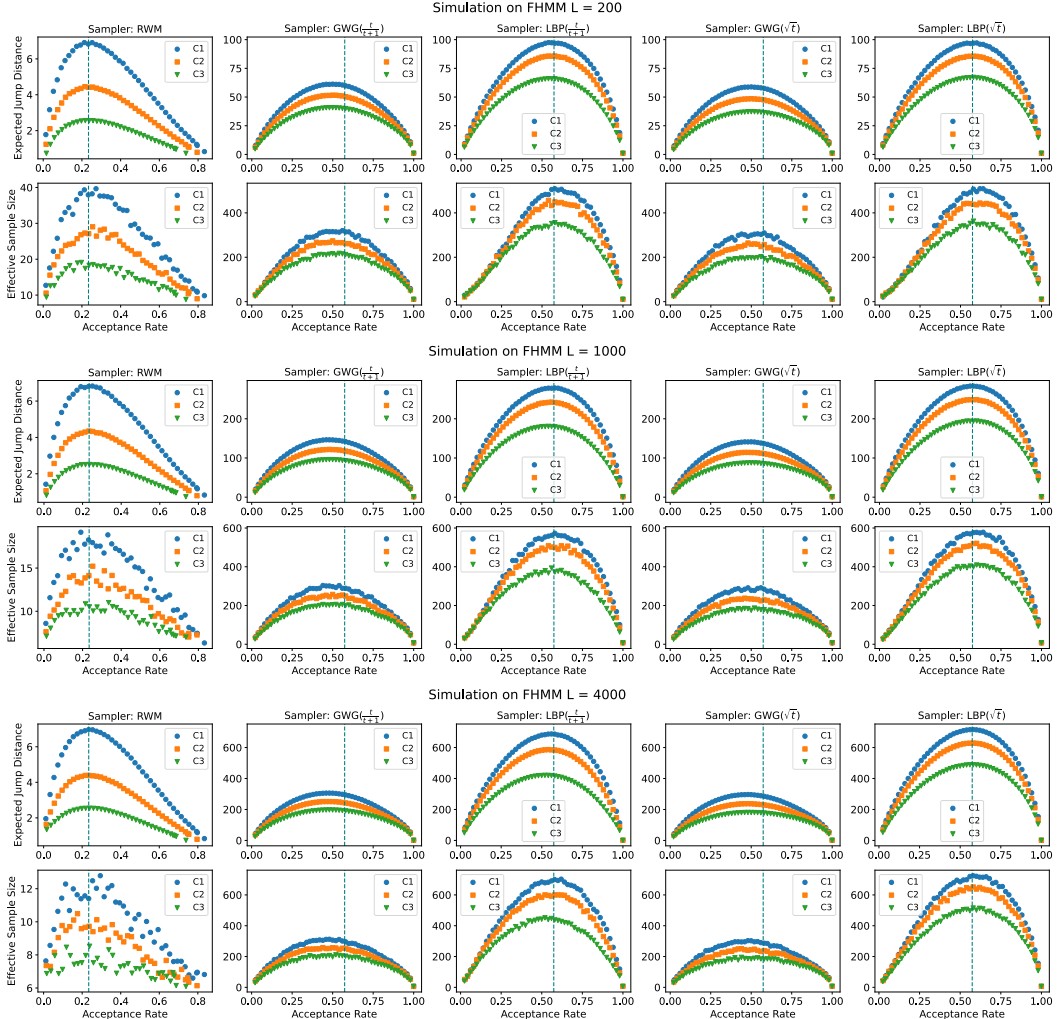

Figure 9: Simulation Results on FHMM

For each configuration, we sample Markov chain $x_{1:40000}$, use $x_{1:20000}$ for burn in, use $x_{20001:40000}$ to estimate expected acceptance rate, expected jump distance, effective sample size.

We give the scatter plot of $(a, d)$ and $(a, r)$ in figure 10. And we examine the performance of our adaptive algorithm in table 14 and table 15.

| Size | $L=200$ | | | $L=1000$ | | | $L=4000$ | | |
|---|---|---|---|---|---|---|---|---|---|
| Sampler | C1 | C2 | C3 | C1 | C2 | C3 | C1 | C2 | C3 |
| RWM-1 | 0.83 | 0.79 | 0.74 | 0.83 | 0.79 | 0.73 | 0.83 | 0.80 | 0.74 |
| ARWM | 6.85 | 4.41 | 2.55 | 6.79 | 4.32 | 2.50 | 6.94 | 4.40 | 2.54 |
| GRWM | 6.91 | 4.45 | 2.56 | 6.83 | 4.35 | 2.51 | 6.97 | 4.39 | 2.54 |
| GWG($\frac{t}{t+1}$)-1 | 1.00 | 1.00 | 1.00 | 1.00 | 1.00 | 1.00 | 1.00 | 1.00 | 1.00 |
| AGWG($\frac{t}{t+1}$) | 59.96 | 50.53 | 40.05 | 142.24 | 118.18 | 93.44 | 294.85 | 243.23 | 191.70 |
| GGWG($\frac{t}{t+1}$) | 61.24 | 51.63 | 40.74 | 146.40 | 121.91 | 95.88 | 305.78 | 251.09 | 197.28 |
| LBP($\frac{t}{t+1}$)-1 | 1.00 | 1.00 | 1.00 | 1.00 | 1.00 | 1.00 | 1.00 | 1.00 | 1.00 |
| ALBP($\frac{t}{t+1}$) | 97.11 | 85.70 | 65.77 | 278.20 | 242.01 | 179.51 | 687.29 | 585.02 | 416.32 |
| GLBP($\frac{t}{t+1}$) | 97.47 | 85.78 | 65.97 | 278.59 | 242.52 | 180.60 | 687.74 | 586.70 | 420.53 |
| GWG($\sqrt{t}$)-1 | 1.00 | 1.00 | 1.00 | 1.00 | 1.00 | 1.00 | 1.00 | 1.00 | 1.00 |
| AGWG($\sqrt{t}$) | 57.52 | 47.55 | 36.53 | 137.11 | 111.40 | 85.78 | 286.03 | 230.52 | 177.22 |
| GGWG($\sqrt{t}$) | 58.83 | 48.53 | 37.21 | 141.15 | 114.24 | 87.45 | 296.62 | 237.35 | 180.79 |
| LBP($\sqrt{t}$)-1 | 1.00 | 1.00 | 1.00 | 1.00 | 1.00 | 1.00 | 1.00 | 1.00 | 1.00 |
| ALBP($\sqrt{t}$) | 96.64 | 85.77 | 66.91 | 283.12 | 248.95 | 193.99 | 715.20 | 627.85 | 488.14 |
| GLBP($\sqrt{t}$) | 97.14 | 85.58 | 67.14 | 283.51 | 248.84 | 194.34 | 716.13 | 628.80 | 488.50 |

Table 11: Expected Jump Distance on FHMM

| Size | $L=200$ | | | $L=1000$ | | | $L=4000$ | | |
|---|---|---|---|---|---|---|---|---|---|
| Sampler | C1 | C2 | C3 | C1 | C2 | C3 | C1 | C2 | C3 |
| RWM-1 | 9.83 | 8.98 | 8.78 | 6.33 | 7.26 | 7.01 | 6.82 | 6.14 | 6.09 |
| ARWM | 35.88 | 28.73 | 18.09 | 18.45 | 13.32 | 10.68 | 10.99 | 10.04 | 8.37 |
| GRWM | 39.65 | 29.04 | 19.09 | 19.15 | 15.22 | 11.00 | 12.79 | 10.49 | 8.52 |
| GWG($\frac{t}{t+1}$)-1 | 10.78 | 10.43 | 9.96 | 7.13 | 6.91 | 7.22 | 6.50 | 5.82 | 6.69 |
| AGWG($\frac{t}{t+1}$) | 306.97 | 262.30 | 213.37 | 288.52 | 245.50 | 196.80 | 295.08 | 241.20 | 195.35 |
| GGWG($\frac{t}{t+1}$) | 320.12 | 273.95 | 217.62 | 303.42 | 257.10 | 205.59 | 312.35 | 257.75 | 210.94 |
| LBP($\frac{t}{t+1}$)-1 | 10.70 | 10.26 | 10.58 | 7.25 | 7.25 | 7.05 | 5.97 | 6.34 | 6.76 |
| ALBP($\frac{t}{t+1}$) | 499.13 | 455.94 | 352.52 | 573.35 | 487.63 | 383.97 | 679.06 | 594.70 | 436.70 |
| GLBP($\frac{t}{t+1}$) | 508.67 | 456.24 | 356.10 | 572.88 | 508.27 | 393.32 | 702.34 | 600.21 | 451.36 |
| GWG($\sqrt{t}$)-1 | 10.30 | 10.22 | 11.06 | 6.69 | 7.90 | 7.09 | 6.53 | 6.72 | 6.80 |
| AGWG($\sqrt{t}$) | 294.38 | 251.57 | 190.81 | 278.17 | 227.18 | 186.24 | 289.14 | 232.77 | 184.94 |
| GGWG($\sqrt{t}$) | 309.79 | 264.26 | 202.46 | 291.43 | 238.06 | 186.76 | 302.98 | 251.14 | 194.88 |
| LBP($\sqrt{t}$)-1 | 9.86 | 10.52 | 10.82 | 6.98 | 7.24 | 7.68 | 6.05 | 6.59 | 6.52 |
| ALBP($\sqrt{t}$) | 502.23 | 443.64 | 348.77 | 575.49 | 524.64 | 406.15 | 727.50 | 645.50 | 504.30 |
| GLBP($\sqrt{t}$) | 508.85 | 444.36 | 362.72 | 578.29 | 520.92 | 408.98 | 724.64 | 651.81 | 515.16 |

Table 12: Effective Sample Size on FHMM

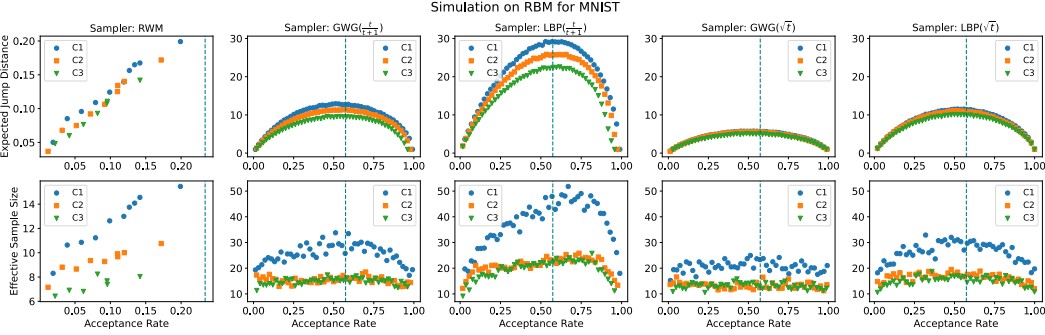

Figure 10: Simulation Results on RBM

| Size | $L = 200$ | | | $L = 1000$ | | | $L = 4000$ | | |
|---|---|---|---|---|---|---|---|---|---|
| Sampler | C1 | C2 | C3 | C1 | C2 | C3 | C1 | C2 | C3 |
| RWM-1 | 136.30 | 129.16 | 30.23 | 58.29 | 58.03 | 61.32 | 112.92 | 112.73 | 110.94 |
| ARWM | 139.53 | 138.36 | 29.94 | 60.08 | 60.02 | 58.61 | 120.83 | 119.95 | 120.38 |
| GRWM | 137.61 | 123.72 | 30.14 | 66.84 | 61.19 | 58.43 | 120.42 | 120.39 | 118.40 |
| GWG($\frac{t}{t+1}$)-1 | 49.54 | 48.86 | 56.84 | 113.86 | 112.89 | 82.67 | 282.79 | 282.29 | 281.53 |
| AGWG($\frac{t}{t+1}$) | 48.80 | 64.40 | 68.73 | 119.27 | 124.07 | 88.57 | 315.22 | 315.52 | 313.65 |
| GGWG($\frac{t}{t+1}$) | 49.76 | 48.60 | 57.24 | 118.77 | 118.59 | 88.57 | 315.45 | 314.77 | 315.27 |
| LBP($\frac{t}{t+1}$)-1 | 53.94 | 69.11 | 75.98 | 129.92 | 134.42 | 91.59 | 295.47 | 294.56 | 292.91 |
| ALBP($\frac{t}{t+1}$) | 43.73 | 57.83 | 64.59 | 92.14 | 129.28 | 93.24 | 315.10 | 327.57 | 308.84 |
| GLBP($\frac{t}{t+1}$) | 57.84 | 57.21 | 63.94 | 136.68 | 140.43 | 100.02 | 315.52 | 314.08 | 309.44 |
| GWG($\sqrt{t}$)-1 | 231.12 | 196.52 | 56.26 | 112.63 | 110.48 | 109.11 | 279.85 | 279.51 | 277.10 |
| AGWG($\sqrt{t}$) | 209.90 | 218.15 | 55.81 | 113.70 | 119.34 | 114.59 | 964.08 | 314.70 | 314.63 |
| GGWG($\sqrt{t}$) | 218.94 | 218.49 | 55.59 | 116.99 | 117.65 | 112.57 | 314.86 | 314.45 | 311.70 |
| LBP($\sqrt{t}$)-1 | 256.23 | 248.33 | 62.95 | 147.23 | 128.65 | 121.14 | 1069.63 | 945.32 | 292.23 |
| ALBP($\sqrt{t}$) | 57.65 | 57.08 | 63.78 | 133.30 | 130.70 | 98.33 | 313.61 | 311.16 | 308.09 |
| GLBP($\sqrt{t}$) | 232.46 | 230.29 | 64.57 | 129.31 | 153.27 | 130.77 | 315.30 | 313.04 | 309.27 |

Table 13: Running Time on FHMM

| Size | $h = 100$ | $h = 400$ | $h = 1000$ |
|---|---|---|---|
| RWM-1 | 0.20 | 0.17 | 0.14 |
| ARWM | 0.19 | 0.17 | 0.14 |
| GRWM | 0.20 | 0.17 | 0.14 |
| GWG($\frac{t}{t+1}$)-1 | 0.99 | 0.98 | 0.96 |
| AGWG($\frac{t}{t+1}$) | 12.75 | 11.31 | 9.62 |
| GGWG($\frac{t}{t+1}$) | 12.91 | 11.36 | 9.58 |
| LBP($\frac{t}{t+1}$)-1 | 0.99 | 0.98 | 0.96 |
| ALBP($\frac{t}{t+1}$) | 29.03 | 26.07 | 22.47 |
| GLBP($\frac{t}{t+1}$) | 29.19 | 25.85 | 22.55 |
| GWG($\sqrt{t}$)-1 | 0.99 | 0.99 | 0.99 |
| AGWG($\sqrt{t}$) | 5.74 | 5.50 | 5.04 |
| GGWG($\sqrt{t}$) | 5.76 | 5.58 | 5.10 |
| LBP($\sqrt{t}$)-1 | 1.00 | 1.00 | 1.00 |
| ALBP($\sqrt{t}$) | 11.41 | 10.65 | 9.93 |
| GLBP($\sqrt{t}$) | 11.53 | 11.09 | 10.10 |

Table 14: Expected Jump Distance on RBM

| Size | $h = 100$ | $h = 400$ | $h = 1000$ |
| --- | --- | --- | --- |
| RWM-1 | 15.46 | 10.76 | 8.04 |
| ARWM | 15.08 | 11.13 | 8.82 |
| GRWM | 15.46 | 10.76 | 8.24 |
| GWG($\frac{t}{t+1}$)-1 | 19.42 | 14.45 | 12.70 |
| AGWG($\frac{t}{t+1}$) | 31.71 | 16.42 | 16.21 |
| GGWG($\frac{t}{t+1}$) | 33.77 | 18.51 | 17.36 |
| LBP($\frac{t}{t+1}$)-1 | 17.99 | 13.38 | 11.16 |
| ALBP($\frac{t}{t+1}$) | 48.20 | 25.59 | 23.61 |
| GLBP($\frac{t}{t+1}$) | 51.82 | 25.83 | 25.78 |
| GWG($\sqrt{t}$)-1 | 21.03 | 13.59 | 12.20 |
| AGWG($\sqrt{t}$) | 21.92 | 13.51 | 15.52 |
| GGWG($\sqrt{t}$) | 24.97 | 16.58 | 15.81 |
| LBP($\sqrt{t}$)-1 | 19.72 | 12.02 | 10.77 |
| ALBP($\sqrt{t}$) | 33.43 | 16.95 | 17.28 |
| GLBP($\sqrt{t}$) | 32.90 | 19.51 | 18.74 |

Table 15: Effective Sample Size on RBM

| Size | $h = 100$ | $h = 400$ | $h = 1000$ |
| --- | --- | --- | --- |
| RWM-1 | 67.37 | 59.54 | 43.48 |
| ARWM | 69.71 | 61.24 | 42.28 |
| GRWM | 67.37 | 59.54 | 44.52 |
| GWG($\frac{t}{t+1}$)-1 | 92.75 | 93.28 | 69.18 |
| AGWG($\frac{t}{t+1}$) | 99.34 | 95.26 | 74.14 |
| GGWG($\frac{t}{t+1}$) | 94.70 | 96.48 | 73.09 |
| LBP($\frac{t}{t+1}$)-1 | 118.60 | 116.04 | 87.54 |
| ALBP($\frac{t}{t+1}$) | 84.03 | 144.03 | 90.43 |
| GLBP($\frac{t}{t+1}$) | 121.71 | 119.38 | 91.12 |
| GWG($\sqrt{t}$)-1 | 109.86 | 94.40 | 69.86 |
| AGWG($\sqrt{t}$) | 94.97 | 94.63 | 72.30 |
| GGWG($\sqrt{t}$) | 96.42 | 94.23 | 72.93 |
| LBP($\sqrt{t}$)-1 | 116.63 | 116.15 | 86.23 |
| ALBP($\sqrt{t}$) | 120.47 | 118.61 | 88.35 |
| GLBP($\sqrt{t}$) | 118.45 | 118.86 | 89.34 |

Table 16: Running Time on RBM