# OpenReview forum: "Optimal Scaling for Locally Balanced Proposals in Discrete Spaces"
_NeurIPS.cc/2022/Conference — NeurIPS 2022 Accept_

### Official Review · Reviewer_GHqz · 2022-07-11

**Rating:** 8
**Confidence:** 4
**Soundness:** 4 excellent
**Presentation:** 3 good
**Contribution:** 3 good

**Summary:**

The optimal scaling problem is the problem of finding the right scale of each jump in Metropolis-Hasting that leads to optimal convergence rate. Even though the problem has been well-studied in the continuous settings, it has been seldomly explored in discrete settings. As a contribution in this direction, the authors study the locally balanced proposal (LBP) proposed by Zanella (2020) which is designed to move one coordinate at a time, with the proposal distribution depending on the likelihood ratio between the local change. Later, Sun et al. (2021) propose a more general version of LBP which can switch multiple coordinates in one step. With this version of LDP, the authors are able to derive the optimal scaling, and with this scaling, they show that the multi-step LBP has an asymptotic acceptance rate that is independent of the target distribution, which is greater than that of the random walk Metropolis.

**Questions:**

As mentioned above, I would like to know to what extent the choice of $\epsilon$ affects the convergence rate in Eq (6).

**Limitations:**

In addition to what the authors suggest in the Discussion, one might try to find (non-asymptotic) convergence rate as a function of $\epsilon$. There is also the question of finding the best weight function $g(t)$ which might depend on the task at hand. Can we find a heuristic on $g$ that gives us optimal or near-optimal performance?

**Strengths And Weaknesses:**

# Strengths
This paper provides sufficient contribution towards efficient Metropolis-Hasting algorithms on discrete spaces. The authors provide thorough analyses and experiments, which agree with the results. I find the application on efficient energy based model to be very interesting.

# Weaknesses
While the paper can be easily read by experts in the fields, I feel that some additions/explanations would be appreciated by outsiders. For example, the authors should explain what **scale** means in both continuous and discrete settings, and they should give a precise definition of **optimal scaling problem** in Section 1 or 2.

The authors have the following assumption on the target distribution $\pi(x) = \prod_{i=1}^N p_i^{x_i} (1-p_i)^{1-x_i}$:
$$ \epsilon < \min (p_j , 1-p_j ) < \frac{1}{2} - \epsilon \qquad \text{for some } \epsilon \in (0,\frac{1}{4}). $$
How the choice of $\epsilon$ in Eq. (2) affects the rate of convergence in Eq. (6) is still unclear to me. I have a hunch that the quality of the Taylor approximation in Lemma 3.3 depends on $\epsilon$ as well. All in all, I think the authors should expand more, in the main paper, on the effect of $\epsilon$ on the efficiency (I see that the choices of $\epsilon$ are mentioned in Appendix C.1 but not in the main paper), and whether the assumption above is reasonable in real-life scenarios (e.g. in EBM).

In the experiments, the authors only compare between two methods, namely LBP and RWM. Even though there are many experiments in the past which have already shown effectiveness of LBP against other methods, such as RWM, Gibbs sampling, the Hamming Ball sampler and continuous relaxation based methods, I still want to see how LBP performs against some of these methods, at least in energy based models.

In addition, I have found some potential mathematical typos in this paper:
* Page 3, Eq (2), $\wedge$ is used here but has never been defined before.
* Page 4, Eq (8), $a(R)R$ should be equal $2lN^{\frac{2}{3}}\Phi\left(-\frac{1}{2}\lambda_1 l ^{\frac{3}{2}}\right)$ in the asymptotic limit, not exactly equal.
* Page 14, Eq (25), what is $M_0$?
* Page 14, Eq (29) and (30), should the middle expressions be put in absolute values?
* Page 14, Eq (29) and (31), $\mathbb{P}(|W(x,u)-W|>N^{\frac{1}{2}})$ should be $\mathbb{P}(|W(x,u)-W|>tN^{\frac{1}{2}})$.
* Page 15, Eq (36), what is the appropriate value of $t$ in order to apply Lemma A.1?
* Page 16, Eq (58), I am not sure how the factor of $1+O(N^{-\frac{1}{3}})$ pops up here.
* Page 18, Eq (84), $X_N$ should be $x_N$, $u_i,\ldots u_R$ should be $u_i,\ldots, u_R$ and $u'_i,u_R$ should be $u'_i,\ldots, u_R$.
* Page 20, Line 529, "we have:" -> "We have:"
* Page 23, Eq (164), should the lower order term depends on $t$ and $\lambda_2$ as well? Also, $R^{\frac{3}{2}}$ should be $R^{\frac{3}{4}}$.

---

> ### Author Response · Authors · 2022-08-02
> **Response to Reviewer GHqz (Part I)**
>
> Thank you for your valuable feedbacks! Please see our responses below:
>
> **Definition of the scale**
>
> Thanks for pointing out our insufficient introduction of the scale.  We have followed your advice and introduced the scale in continuous space and in discrete space in section 1 in our revision (marked as blue). Due to the space limit, we will provide formal discussion in the appendix.
>
> **How $\epsilon$ influences the convergence rate?**
>
> Thank you for raising this interesting question. We have added the following analysis in our revision.
> The convergence of Eq. (6) does not depend on the value of $\epsilon$ in Eq. (2).
> Based on the proof above, we can know Eq.(6) converges at the rate $O(N^{-\frac{1}{12}})$. But the convergence of the optimal acceptance rate depends on the $\epsilon$. We can first consider two extreme cases for intuition. When all $p_i$ are close to $\frac{1}{2}$, $\lambda_1$ in Eq.(7) will be close to $0$ and the optimal acceptance rate will be close to $1$; when all $p_i$ are close to $0$ or $1$, $\lambda_1$ in Eq. (7) will be close to $\infty$ and the optimal acceptance rate will be close to $0$. Hence, the main purpose to use fixed $\epsilon$ is to give upper and lower bounds for $\lambda_1$ in Eq. (7) and $\lambda_2$ in Eq. (21), such that the optimal acceptance rate can converge to $0.574$ and $0.234$ as in Corollary 3.2 and Corollary 3.9.
>
> Next, we discuss how the model dimension $N$ in Eq. (1) needed in terms of $\epsilon$ to make sure the optimal convergence to $0.574$. When all $p_i$ have the extreme value determined by $\epsilon$, we can consider the following two situations:
>
> * All $|p_i - 0.5| = \epsilon \rightarrow 0$. Then we have:
> \begin{equation}
> \lambda_1^2 \approx \frac{\sum_{i=1}^N 0.5 \cdot 4 \epsilon^2}{4 \cdot \sum_{i=1}^N 0.5} = \epsilon^2
> \end{equation}
> When the expected acceptance is $0.574$, we need $\lambda_1 l^\frac{3}{2} = O(\epsilon l^\frac{3}{2})$ equals to a constant, which means $l$ has the same order as $\epsilon^{-\frac{2}{3}}$. Since we have $R = l N^\frac{2}{3} \le N$, we need $\epsilon^{-\frac{2}{3}} = O(N^\frac{1}{3})$. As a result, we requires $\epsilon^{-1} = O(N^{\frac{1}{2}})$, which basically means we need $N \ge \epsilon^{-2}$ to have the optimal acceptance rate converges to $0.574$.
>
> * All $0.5 - |p_i - 0.5| = \epsilon \rightarrow 0$. We have:
> \begin{equation}
>     \lambda_1^2 \approx \frac{\sum_{i=1}^N \epsilon \epsilon^{-2}}{4 \epsilon^2 \sum_{i=1}^N \epsilon} = \frac{1}{4}\epsilon^{-4}
> \end{equation}
> When the expected acceptance is $0.574$, we need $\lambda_1 l^\frac{3}{2} = O(\epsilon^{-2} l^\frac{3}{2})$ equals to a constant, which means $l$ has the same order as $\epsilon^{\frac{4}{3}}$. Since we have $R = l N^\frac{2}{3} \ge 1$, we have $l^{-1} = O(N^\frac{2}{3})$.  As a result, we requires $\epsilon^{-1} = O(N^{-\frac{1}{2}})$, which basically means we need $N \ge \epsilon^{-2}$ to have the optimal acceptance rate converges to $0.574$.
>
> So, both situations show that we need $N \ge C' \epsilon^{-2}$, for some constant $C'$, to make sure the optimal acceptance rate converges to $0.574$.
>
> (RESPONSE TO BE CONTINUED IN PART II)
>
> >[1] Sansone, Emanuele. "Lsb: Local self-balancing mcmc in discrete spaces." International Conference on Machine Learning. PMLR, 2022. \
> [2] Grathwohl et. al, “Oops I Took A Gradient: Scalable Sampling for Discrete Distributions”, ICML 2021 \
> [3] Du et.al, “Implicit Generation and Modeling with Energy-Based Models”, NeurIPS 2019

---

> > ### Comment · Reviewer_GHqz · 2022-08-09
> > **Reply**
> >
> > Does your experiment suggest that we can do better than $O(\epsilon^{-2})$?

---

> > > ### Author Response · Authors · 2022-08-09
> > > **Experiments results for $N = O(\epsilon^{-2})$**
> > >
> > > We first want to confess that in our previous reply, we made a trivial error in the second situation in estimating $\lambda_1^2$. In particular, when all $p_i = \epsilon \rightarrow 0$ and use locally balanced function $g(t) = \sqrt{t}$, we have:
> > > \begin{equation}
> > >     \lambda_1^2 = \frac{\sum_{j=1}^N \epsilon \sqrt{\frac{1 - \epsilon}{\epsilon}} (\sqrt{\frac{1 - \epsilon}{\epsilon}} - \sqrt{\frac{\epsilon}{1 - \epsilon}})^2 }{4 (\sqrt{\epsilon(1-\epsilon)})^2 \sum_{i=1}^N \sqrt{\epsilon (1 - \epsilon)} } \approx \frac{\sum_{j=1}^N \epsilon^\frac{1}{2} \epsilon^{-1} }{4 \epsilon \sum_{j=1}^N \epsilon^{-\frac{1}{2}}} = \frac{1}{4}\epsilon^{-2}
> > > \end{equation}
> > > So, the second situation only requires $N = O(\epsilon^{-1})$.
> > >
> > > We conduct extra numerical simulations to verify our results. To simplify the experiments, we assume all dimensions have the same configurations: $p_i = p$. For different $p$, we report the size of $N$ needed to guarantee that the optimal acceptance rate is $0.574$.
> > >
> > > * Situation 1: $p = 0.5 - \epsilon$
> > > | $\epsilon$ | 0.1 | 0.05 |  0.025 | 0.0125 |
> > > |:-----------:|:--------:|:--------:|:--------:|:--------:|
> > > | N | 10 | 40  | 160 | 640 |
> > >
> > > * Situation 2: $p = \epsilon$
> > > | $\epsilon$ | 0.01 | 0.005 |  0.0025 | 0.00125 |
> > > |:-----------:|:--------:|:--------:|:--------:|:--------:|
> > > | N | 50 | 100  | 200 | 400 |

---

> > > > ### Comment · Reviewer_GHqz · 2022-08-09
> > > > **Reply**
> > > >
> > > > Thanks for the additional experiments. I am satisfied with all the answers.

---

> ### Author Response · Authors · 2022-08-02
> **Response to Reviewer GHqz (Part II)**
>
> (RESPONSE PART II CONTINUED HERE)
>
> **“...how LBP performs against some of these methods, at least in energy based models…”**
>
> We have followed the reviewer’s advice to compare against some other samplers in the scenario of energy based model learning, using the dynamic-MNIST setting as described in our paper. We only consider methods that are applicable in discrete space, as learning EBMs in continuous space is much easier [3].
>
> | Methods | # MH steps for PCD | Time per EBM update |  log-likelihood |
> |:-----------:|:--------:|:--------:|:--------:|
> | Rand-Walk | 2000 | 12.86s  | -86.46 |
> | HammingBall | 500 | 18.85s | -95.61 |
> | Gibbs [2] | 800 | 9.5s | -121.19 |
> | ALBP (ours) | 20 | 0.5s | -79.32 |
>
> The results of Gibbs are obtained from [2] since the same GPU and pytorch framework are used. Overall we can see that to stabilize the EBM training, the baseline samplers would require hundreds or even thousands of PCD steps per gradient update, while also achieving much worse likelihood after 24h of training. Using the adaptive steps with our derived optimal acceptance rate, the LBP sampler can be much faster and achieve much better results in only 7h.
>
> **Typos.**
>
> Thanks for pointing out the typos in our work, especially for those in the long proof in the appendix. We really appreciate it and we have fixed them in our revision.
>
> **“... find a heuristic on  $g$...”**
>
> Choosing the right $g$ is not the focus of our paper, and our derived optimal scaling is independent of this choice. Nevertheless, how to choose $g$ is still an interesting open question.  Empirically, people find the four choices $g(t) = \sqrt{t}, \frac{t}{t+1}, \max\{t, 1\}, \min\{t, 1\}$ have comparably good performance, and $g(t) = \sqrt{t}$ is best among them in most of the cases. There is a recent work [1] that tries to learn the weight function $g$. [1] considers learning the best linear combination of the four choices mentioned above. Empirically, learning the weight function does not bring too much improvement [1].
>
> >[1] Sansone, Emanuele. "Lsb: Local self-balancing mcmc in discrete spaces." International Conference on Machine Learning. PMLR, 2022. \
> [2] Grathwohl et. al, “Oops I Took A Gradient: Scalable Sampling for Discrete Distributions”, ICML 2021 \
> [3] Du et.al, “Implicit Generation and Modeling with Energy-Based Models”, NeurIPS 2019

---

> > ### Comment · Reviewer_GHqz · 2022-08-09
> > **Reply**
> >
> > Impressive results. What about the memory requirement compared to the other methods?

---

> > > ### Author Response · Authors · 2022-08-09
> > > **Memory**
> > >
> > > We updated the comparison table with an extra column for GPU memory usage. Overall ALBP (ours) enjoys similar memory usage as other methods. Note that the memory usage of baseline methods might be dependent on the implementation. Although methods like Gibbs sampling need multiple times of energy function evaluation per MH step, it can be serialized so the memory consumption can be constant (though one can make trade-off between memory and parallelism/speed for Gibbs sampling).
> > >
> > > | Methods | # MH steps for PCD | Time per EBM update |  log-likelihood | GPU Memory |
> > > |:-----------:|:--------:|:--------:|:--------:|:--------:|
> > > | Rand-Walk | 2000 | 12.86s  | -86.46 | 2005M |
> > > | HammingBall | 500 | 18.85s | -95.61 | 2879M |
> > > | Gibbs [2] | 800 | 9.5s | -121.19 | 1971M |
> > > | ALBP (ours) | 20 | 0.5s | -79.32 | 1995M |

---

### Official Review · Reviewer_xSRR · 2022-07-11

**Rating:** 6
**Confidence:** 4
**Soundness:** 2 fair
**Presentation:** 3 good
**Contribution:** 3 good

**Summary:**

This paper studies the optimal scaling of Metropolis-Hastings algorithms on the cube $\mathbb{Z}_2^N$. The analysis of optimal scaling for Markov chain Monte Carlo methods has been studied extensively for continuous state spaces, but not for discrete spaces. The authors provide a scaling analysis for the locally balanced proposal (LBP) in [2]. The authors show that the LBP has a better asymptotic property than the random-walk Metropolis (RWM) algorithm. Simulation results support the theoretical results.

**Questions:**

- Does Theorem 3.1 hold regardless of the choice of $g$? What is the optimal choice of $g$ that minimises the value of $\lambda_1^2$?

**Limitations:**

The main limitation is the gap between theory and claim. The authors evaluate the expected squared jump distance. However, the meaning of this statistic is not clear in the current context, since the limit process is a pure jump process and not the Langevin diffusion.

**Strengths And Weaknesses:**

The results in this paper are interesting and show a systematic difference between LBP and RWM. LBP is a new and useful method for discrete state space. I believe that the results in this paper are a good contribution to this field if the following weak points are solved. From the theoretical point of view, the result shows the similarity of LBP with the Langevin algorithm. Therefore, it is reasonable that LBP is superior to RWM. From a practical point of view, the result provides a guide for the choice of tuning parameters.

However, I feel that there is a gap between theory and claim. The efficiency presented in this paper only makes sense if the limit is a class of diffusion processes. However, the literature suggests that the limit is not a diffusion process, but a  Poisson process. Therefore, the meaning of efficiency is not obvious to me in this paper.

Also, I have a less clear picture of comparing LBP and RWM. The analysis of LBP focuses on the case $p\ll 1/2$, while the analysis of RWM focuses on the case $p \approx 1/2$. Therefore, it isn't easy to draw a general conclusion comparing the two algorithms. However, the paper [1] is useful in this comparison.

As a novelty, I need to mention that the scaling limit for discrete space is not new. More precisely, the LBP results are new, but the RWM results are known. See [1].  In [1], only the iid target distribution is considered, so the class of target distribution in this paper is much more general in this sense.  However, the authors only study the case where $p \approx 1/2$, but [1] also studied $p \ll 1/2$ and identified the limit process.

The paper has good potential, but I think there is room for improvement.

Minor
- I recommend checking statements and proofs once again. There are many minor errors. For example, in Lemma 3.3 $A+B$ is a limit of the random variable, say $X_N$, but $A+B$ still depends on $N$.
- It would be nice if the authors could comment on the guideline for choosing the function $g$.

[1] Gareth O. Roberts (1998) Optimal metropolis algorithms for product measures on the vertices of a hypercube, Stochastics and Stochastic Reports, 62:3-4, 275-283

[2] Haoran Sun, Hanjun Dai, Wei Xia, and Arun Ramamurthy. Path auxiliary proposal for mcmc in discrete space. In International Conference on Learning Representations, 2021

(I updated my rating after reading the authors response and follow-up.)

---

> ### Author Response · Authors · 2022-08-02
> **Response to Reviewer xSRR**
>
> Thank you for your valuable feedbacks! Please see our responses below:
>
> **“...the gap between theory and claim. The efficiency only makes sense if the limit is a class of diffusion process...”**
>
> Thank you for pointing out the potential gap. Allow us to provide detailed explanations for why EJD can still be the correct metric to evaluate the efficiency of samplers in discrete spaces.
>
> We can consider a more general case where the target distribution is the product of categorical distributions:
> $$ \pi(x) = \prod_{i=1}^N \pi_i(x_i) $$
> where $\pi_i(x_i)$ is a categorical distribution for $\{1, ..., n\}$. Let the LBP chain, with $R = lN^\frac{2}{3}$, being denoted as $\{x^{(1)}, x^{(2)}, ...\}$. We consider the related one-dimensional process
> $$Z^N_t = x_1^{\\lfloor tN^\\frac{1}{3} \\rfloor}$$
> We know that when $N$ is large enough, $Z^N_t$ converges to a jump process with the generator $\lambda(l) Q$, such that
> $$Q_{ij} = [-\sum_{k\neq i} g(\frac{\pi_1(k)}{\pi_1(i)})] 1_{j = i} + g(\frac{\pi_1(j)}{\pi_1(i)}) 1_{j\neq i}
> $$
> where $g(\cdot)$ is the locally balanced function, and $\lambda(l)$ is a scalar determined by $l$. When we say $Z^N_t$ is determined by the generator $\lambda(l)Q$, we means the distribution $\rho(t)$ for $Z^N_t$ satisfies
> $$ \rho(t+h) = \rho(t) \exp(\lambda(l)Qh) $$
>
> The reason for $Q$ has this expression is that when $N$ is large enough, the logarithm of the acceptance ratio (the ratio before clip by 1) has standard deviation $O(N)$ (see Lemma 3.6). Hence, the acceptance rate is almost independent with the first dimension of the current state and the generator should give the same transition probability as the LBP.
>
> For any test functions $f(\cdot)$, the inverse auto-correlation of the jump process is proportional to $\lambda(l)$. Also, as a jump process, we know $\lambda(l)$ is proportional to the expected jump distance (EJD). This is different from the diffusion process in continuous space, whose velocity is characterized by ESJD. This is the reason why we should use EJD instead of ESJD in discrete space.
>
> We have added the discussion above in Appendix B.1.
>
> **Assumption for RWM and the novelty for the results of RWM.**
>
> * We want to clarify that the main purpose of this work is to find the optimal scaling of LBP. The reason for deriving the scaling of RWM is merely to compare LBP and RWM at their optimal scaling.
> * To derive the optimal acceptance rate of RWM, it is inevitable to assume that $p$ is close to 0.5. Without this assumption, the number of sites to flip per step (which is denoted as $R$ in our work) is $O(1)$, which prevents the use of the central limit theorem to estimate the acceptance rate. Hence, both our work and the limit process in [2] consider the case when $p$ is close to $0.5$.
> * Even without the assumption that $p$ is close to 0.5, the number of sites RWM should flip per step is still $O(1)$, and LBP is still $O(N^\frac{2}{3})$ more efficient than RWM. We will clarify this in the revision.
>
> **Improper statement in Lemma 3.3**
>
> Thank you for pointing out this issue! We have rephrased Lemma 3.3 from “sum … weakly converges to $A+B$” to “sum … - (A+B) converges to 0” in our revision.
>
> **“...comment on the guideline for choosing the function $g$”**
>
> Choosing the right $g$ is not the focus of this paper, and the derived optimal scaling is independent of this choice. Nevertheless, how to choose $g$ is still an interesting open question.  Empirically, the four choices $g(t) = \\sqrt{t}, \\frac{t}{t+1}, \\max\\{t, 1\\}, \\min\\{t, 1\\}$ have been found to have comparably good performance, and $g(t) = \\sqrt{t}$ is best among these in most of the cases. There is a recent work [3] that tries to learn the weight function $g$. [3] considers learning the best linear combination of the four choices mentioned above. Empirically, learning the weight function does not bring much improvement [3].
>
>
> >[1] Gareth O Roberts and Jeffrey S Rosenthal. Optimal scaling for various metropolis-hastings algorithms. Statistical science, 16(4):351–367, 2001. \
> [2] Gareth O Roberts. Optimal metropolis algorithms for product measures on the vertices of a hypercube. Stochastics and Stochastic Reports, 62(3-4):275–283, 1998. \
> [3] Sansone, Emanuele. "Lsb: Local self-balancing mcmc in discrete spaces." International Conference on Machine Learning. PMLR, 2022.

---

> > ### Comment · Reviewer_xSRR · 2022-08-08
> > **Brief reply to Response to Reviewer xSRR**
> >
> >
> > #### Comment on efficiency
> >
> > Appendix B.1 has clearly answered my main question about efficiency. I am still not completely satisfied with this point, but overall I am happy with the answer.
> >
> > What I am not satisfied with is that the results in Appendix B.1 are not rigorous at all. There is neither a formal statement nor a proof. Moreover, the expression of the quantity $\lambda(l)$ is not presented in the appendix. I recommend writing a formal statement with the full expression of $\lambda(l)$ if the expression is not super complicated.
> >
> > #### Comment on the assumption on $p$
> >
> > The authors' comment partially answered my question about the incompatibility of the assumptions for LBP and RWM, but I still have a less clear picture of the comparison of the two. The probability $0.574$ for LBP is for the case $p\ll 1/2$, and $0.234$ for RWM is for the case $p\approx 1/2$. I guess that the LBP result for $p\ll 1/2$ cannot simply be applied to the case $p\approx 1/2$ and the RWM result for $p\approx 1/2$ cannot simply be applied to the case $p\ll 1/2$.  Also, one of the main conclusion, $O(N^{2/3})$ efficiency, is not for Theorem 3.8, but for the brief discussion at the beginning of Section 3.5. It would have been nice if the authors had made more effort to find a formal result about RWM under (2).
> >
> > Minor comment: Strictly speaking, the set (19) is an empty set, since the set (2) and the set of $\pi^{(N)}$ satisfying the last condition in (19) are disjoint.

---

> > > ### Author Response · Authors · 2022-08-08
> > > **Follow up (Part I)**
> > >
> > > Thank you for your reply, we really appreciate your effort! We have followed your advice to provide proofs and refined the paper accordingly. Please kindly see our further response below:
> > >
> > > **No Formal statement of the EJD in Appendix B.1**
> > >
> > > We added a sketch of the proof in Appendix B.1. We will polish the appendix further and make it more rigorous. We hope this would provide an understanding of why using EJD to measure the efficiency of samplers in discrete space.
> > >
> > > To simplify the derivation, we first consider the distribution
> > > $$
> > >     \pi^{(N)}(x) = \prod_{i=1}^N \pi_i(x_i) = \prod_{i=1}^N p^{x_i}(1-p)^{1-x_i}
> > > $$
> > > We can notice that, compared to the target distributions considered in the main text eq (1), we assume the target distribution is identical in each dimension.
> > >
> > > Let the LBP chain, with $R = lN^\frac{2}{3}$, being denoted as $\{x(1), x(2), ... \}$. Since all dimensions are identical, we only need to focus on the first dimension. Denote $w_1 = g(\frac{\pi_1(x_1=0)}{\pi_1(x_1=1)})$ and $w_0 = g(\frac{\pi_1(x_1=1)}{\pi_1(x_1=0)})$. From Lemma. A.1, we can see that:
> > > $$
> > >     \lim_{N\rightarrow \infty} \frac{\mathbb{P}(u, \exists u_j = 1| x_1 = 0)}{\mathbb{P}(u, \exists u_j = 1| x_1 = 1)} = \frac{w_0}{w_1}
> > > $$
> > > That's to say, the probability ratio of $x_1=0$ and $x_1=1$ being flipped equals their weight ratio. Then we compare the acceptance rate in the M-H test. From the proof of the main theorem 3.1, we know the acceptance rate is determined by the term $A$ defined in eq (11)
> > > $$
> > >     A = \frac{1}{W}\sum_{r=1}^R (R-r+1)w_{u_i}(x_{u_i}) - r w_{u_i}(y_{u_i})
> > > $$
> > > We can see that, when the first dimension is flipped in the proposal, the difference of $A$ is $O(N^{-\frac{1}{3}})$ for $x_1=0$ and $x_1=1$. As a result, we have:
> > > $$
> > >     \lim_{N\rightarrow \infty} \frac{\mathbb{P}(\text{accept }|u, \exists u_j=1,  x_1 = 0)}{\mathbb{P}(\text{accept }|u, \exists u_j=1, x_1 = 1)} = 1
> > > $$
> > > Now, we consider the one-dimensional process $Z^N_t = x_1(\lfloor tN^\frac{1}{3}\rfloor)$.
> > > The identical assumption implies that the frequency for a site, for example the first dimension, being selected is $l N^{-\frac{1}{3}}$.
> > > We can easily see that when $N$ is large enough, $Z^N_t$ converges to a jump process $Z_t$, whose generator we denote.
> > > $$Q = \\left[
> > >     \\begin{array}{rr}
> > >     -Q_{01} & Q_{01} \\\\
> > >     Q_{10}   & -Q_{10}
> > >     \\end{array}
> > >     \\right]$$
> > > From the derivation above, we know that
> > > $$
> > > \frac{Q_{01}}{Q_{10}} = \lim_{N\rightarrow \infty} \frac{\sum_u E_{x_{2:N}}[P(u, \exists u_j=1| x_1 = 0) P(\text{accept }|u, \exists u_j=1,  x_1 = 0)]}{\sum_u E_{x_{2:N}}[P(u, \exists u_j=1| x_1 = 1) P(\text{accept }|u, \exists u_j=1,  x_1 = 1)]} = \frac{w_0}{w_1}
> > > $$
> > > Since the sketch of proof above shows that the ratio is independent with the parameter $l$, we have the following important decomposition
> > > $$Q = \lambda(l) Q(p)$$
> > > where $Q(p)$ is a matrix only depending on $p$ and the locally balanced function $g$ selected, and $\lambda(l)$ is a scalar only depending on the parameter $l$.
> > >
> > > Since $Q(p)$ only depends on the target distribution, for any test functions $f(\cdot)$, the inverse auto-correlation of the jump process is proportional to $\lambda(l)$. When we tune $l$, the coefficient $\lambda(l) = l \cdot 2 \Phi(- \lambda_1 l^\frac{3}{2})$ is the multiplication of the proposal frequency and the acceptance rate. The value $\lambda_1$ is defined in eq (7). As a jump process, we don't have to analytically compute the value of $\lambda(l)$, as $\lambda(l)$ is proportional to the expected jump distance (EJD). So, we can tune $l$ by maximizing the EJD, without having to know the formulation of the target distribution.
> > >
> > >
> > > Remark 1: The jump process in discrete space is different from the diffusion process in continuous space. For the diffusion process, the ESJD characterizes its velocity. But for the jump process, the EJD characterizes its velocity. That's why Langevin algorithms tune the step size via ESJD, but our LBP tunes the path length via EJD.
> > >
> > > Remark 2: To simplify the derivation, we assume that the target distributions have identical marginals. For target distributions with non-identical marginal distribution, different dimensions $i = 1, ..., N$ can have different velocity $\lambda_i(l)$. But the sampling process will still converge to the jump process, and we shall still use EJD to measure the efficiency.

---

> > > ### Author Response · Authors · 2022-08-08
> > > **Follow up (Part II)**
> > >
> > > **Efficiency of LBP when $p \approx \frac{1}{2}$**
> > >
> > > The convergence of Eq. (6) does not depend on the value of $\epsilon$ in Eq. (2).
> > > Based on the proof above, we can know Eq.(6) converges at the rate $O(N^{-\frac{1}{12}})$. But the convergence of the optimal acceptance rate depends on the $\epsilon$. We can first consider two extreme cases for intuition. When all $p_i$ are close to $\frac{1}{2}$, $\lambda_1$ in Eq.(7) will be close to $0$ and the optimal acceptance rate will be close to $1$; when all $p_i$ are close to $0$ or $1$, $\lambda_1$ in Eq. (7) will be close to $\infty$ and the optimal acceptance rate will be close to $0$. Hence, the main purpose to use fixed $\epsilon$ is to give upper and lower bounds for $\lambda_1$ in Eq. (7), such that the optimal acceptance rate can converge to $0.574$ as in Corollary 3.2
> > >
> > > When $p \approx \frac{1}{2}$, the optimal acceptance rate can be larger than $0.574$, which makes the LBP algorithm even more efficient. Please see more convergence analysis in Appendix B.2.
> > >
> > > **Efficiency of RWM under eq (2)**
> > >
> > > We prove the efficiency of RWM under eq (2) is $O(1)$ in Appendix B.3.
> > >
> > > When we assume the target distribution belongs to eq (2), the derivation of the optimal acceptance rate $0.234$ is no longer valid. But we can still show the optimal scale is $R = O(1)$ by proving the acceptance rate decreasing exponentially fast.
> > >
> > > In particular, assume we use $R = l N^\beta$ in RWM. Then the acceptance rate can be written as:
> > > $$
> > >     A = \min\{1, A' = \frac{\pi(y)}{\pi(x)} = \frac{\prod_{j=1}^R \pi_{u_j}(y)}{\prod_{j=1}^R \pi_{u_j}(x)}\}
> > > $$
> > > Consider the martingale $M_j, j=0, 1, ..., R$, such that $M_0 = 0$ and
> > > $$
> > >     M_j - M_{j-1} =  \log \frac{\pi_{u_j}(y)}{\pi_{u_j}(x)} - \mathbb{E}[\log \frac{\pi_{u_j}(y)}{\pi_{u_j}(x)}|u_{1:j-1}] = (1 - 2x_{u_j}) \log \frac{p_{u_j}}{1 - p_{u_j}}
> > > $$
> > > By assumption in eq (2), we know that
> > > \begin{align}
> > >     \mathbb{E}[\log \frac{\pi_{u_j}(y)}{\pi_{u_j}(x)}|u_{1:j-1}]
> > >     & = \mathbb{E} [(1 - 2x_{u_j}) \log \frac{p_{u_j}}{1 - p_{u_j}}|u_{1:j-1}] = (1 - 2p_{u_j}) \log \frac{p_{u_j}}{1 - p_{u_j}} \\\\
> > >     & \le 2\epsilon \log \frac{1 - 2 \epsilon}{1 + 2 \epsilon} < 0
> > > \end{align}
> > > And we have
> > > $$
> > >     |M_j - M_{j-1}| \le 2 \left|(1 - 2 \epsilon) \log \frac{\epsilon}{1 - \epsilon}\right| := K
> > > $$
> > > By Azuma-Hoeffding lemma, we have
> > > $$
> > >     \mathbb{P}(|\sum_{j=1}^R \log \frac{\pi_{u_j}(y)}{\pi_{u_j}(x)} - \mathbb{E} [\log \frac{\pi_{u_j}(y)}{\pi_{u_j}(x)}] | \ge R^\frac{3}{4} t) \le 2 e^{\frac{-R^\frac{1}{2} t^2}{K^2}}
> > > $$
> > > For $\beta > 0$, $R$ increases to infinity when $N \rightarrow \infty$. In this case, $\log A'$ concentrates to a value $T \le R\cdot 2 \epsilon \log \frac{1 - 2\epsilon}{1 + 2\epsilon}$ and $A'$ decreases exponentially fast. Hence, the optimal scaling of RWM is $O(1)$.
> > >
> > > **Minor: set in eq (19)**
> > >
> > > Thank you for pointing out our typo. The set in (19) does not have to be a subset of the set in eq (2). We have fixed the typo in our revision.

---

### Official Review · Reviewer_Lua3 · 2022-07-14

**Rating:** 6
**Confidence:** 4
**Soundness:** 3 good
**Presentation:** 2 fair
**Contribution:** 3 good

**Summary:**

The paper provides an optimal scaling analysis of Locally Balanced Proposals (LBP), a recently-proposed class of informed MCMC algorithms for discrete spaces, see in particular [9,10,11]. In particular the paper:

a) Provides an optimal scaling result (theorem 3.1) for LBP on discrete spaces, more specifically on product-form binary spaces. Interestingly, the optimal acceptance rate and asymptotic scaling is the same as MALA on continuous spaces, i.e. an optimal acceptance rate of 0.574 and O(N^1/3) steps for each effective sample in dimension N, see Corollary 3.2.

b) Provides an optimal scaling result for RWM (theorem 3.8) for random-walk metropolis (RWM), under the stronger assumption that each bit has probability close to 0.5. Under this somewhat more artificial scenario, one recovers again the same behavior as RWM in continuous spaces, namely an optimal acceptance rate of 0.234 and O(N) steps for each effective sample in dimension N, see Corollary 3.9. This result seems to be mainly interesting to have a comparison with the one in a).

c) Implements adaptive MCMC versions of multi-step LBP, exploiting the theoretical results of (a) and (b), and tests the resulting algorithms on a variety of sampling tasks in high-dimensional discrete spaces.

d) The proposed methods are applied to energy-based models (EBM) training, obtaining a 2- to 5-fold speedup compared to non-tuned single-step LBP.

**Questions:**

- You use EJD as a metric. It may be helpful to discuss and justify that in a more convincing way. For example, in continuous spaces, the ESJD is typically used. I believe the EJD makes more intuitive sense in discrete spaces than ESJD, but it may be worth discussing if that is the case or not. Also, are the optimality results sensitive to the choice? They seem to be, which makes it important to justify the choice. Also, the sentence in line 98 about ergodic averages being bad since they depend on the specific test function is not that convincing, since also the choice of EJD implies basically a (similarly arbitrary) choice of metric (and implicitly test function?).

- It would help to provide algorithmic box (in supplement) of the actual algorithm you use in simulations, including both the use of gradient approximation and clarifying that the gradient is not recomputed R times (as it is in the methodology of Sun et al (2021)?)

- In the specific setting of Theorem 3.1, the acceptance rates of multi-step LBP depends only on the ratio of normalizing constants Z. Would this still be the case in more realistic cases with non-product targets and gradient approximations? If not, what is your intuition for the results extrapolating well to more realistic scenarios?

- Why are the ESS results in Table 1 not that satisfying for RBM compared to the EJD and to the other models considered?

- Lemma 3.3: I find it hard to understand the exact mathematical statement contained therein (e.g. specify distribution of x and u; which limit are we taking exactly when we say "weakly converges to A+B"? e.g. are we taking a limit where N,R\to\infty? If yes, how can the limit A and B depend on R?). Also, I would remove "Using Taylor's series" since the Lemma should contain the mathematical statement, not how it is proved. Looking at the proof it seems more appropriate to state something like "the difference sum.... -(A+B) converges to 0", rather than "sum... converges to A+B" since A+B depend itself on N and R (like you stated e.g. in (6) of Theorem 3.1).

The paper contains various typos or inaccuracies, e.g.
- line 105: (??)
- line 118: additional "." at the end of line
- line 117: R should be an integer, while l N^(2/3) will not be n general.
- line 140: "we first determines"
- line 185: do Andrieu and Thoms [19] recommend freezing tuning parameters after warmup?
- line 287: typo in "wight"

**Limitations:**

The authors have adequately addressed the limitations and potential negative societal impact of their work.

**Strengths And Weaknesses:**

Strengths:

- there is a large and well-established optimal scaling literature for MCMC in high-d continuous spaces. On the contrary, providing an analogous and meaningful analysis for discrete spaces is not easy and could be helpful to advance the practical and theoretical understanding of informed MCMC in high-d discrete spaces. This paper provides an interesting and significant step in this direction, coupling the optimal scaling framework with LBP and providing a meaningful and non-trivial analysis. Interestingly, the author obtain results that mirror the ones of continuous spaces both in terms of optimal acceptance rates and scaling with dimensionality.

- While recent work using LBP for discrete spaces sampling, e.g. [9,10,11], have witnessed promising empirical results in various tasks including EBM training, there is not much theory (even with some heuristics or on toy cases) quantifying the potential improvements given by LBP in high-dimensions. The analysis of the current paper provides some interesting results in this direction suggesting an O(N^2/3) gain, at least under some simplifying assumptions.

- The numerical results are well-designed, match with remarkable fidelity what predicted by the theory and suggest that the adaptive multi-step LBP schemes under consideration are promising algorithms for discrete space sampling and EBM training.

Weaknesses:

- While the result for LBP is interesting and genuinely novel, the one for RWM is derived under the unsatisfactory assumption that flipping probabilities are close to 0.5 (i.e. the target is close to flat) and is quite reminiscent of (though technically different from) the one in [30], both in terms of setting, spirit and conclusion.

- The results are derived for the specific case of product-form binary targets, which is quite specific and potentially restrictive. Also it somehow feels that in the context of discrete spaces, which is probably harder to study in terms of the questions considered in the paper, specific assumptions about the algorithm and the target distribution can play a more crucial role than in continuous spaces (e.g. use of gradient approximation or not, use of continuous versus discrete time LBP scheme, binary versus more general spaces, sampling bits with or without replacement, etc). For example, it would be interesting to have theoretical results that also quantify the impact of the gradient approximation as in [10] (since without that the cost per iteration of LBP and RWM is not comparable). This however would require considering more complex target distributions.

- Some mathematical statements (e.g. Lemma 3.3) are hard to follow and not rigorous enough. Also, there are quite a few typos throughout the paper. More details below.

- The novelty of the proposed methodology is relatively limited (basically an adaptive version of previously proposed schemes). In that sense the main contribution is the theoretical one, which provides results on optimal acceptance rates to guide tuning, and results quantifying the scaling with dimensionality under some simplifying assumptions.

---

> ### Author Response · Authors · 2022-08-02
> **Response to Reviewer Lua3**
>
> Thank you for your valuable feedbacks! Please see our responses below:
>
> **Assumption for RWM**
>
> We followed the same assumption used in [1].  Without this assumption, the number of sites to flip per step is $O(1)$ and one cannot use the central limit theorem to estimate the acceptance rate for RWM. We clarify that the focus of this paper is mainly on optimal scaling for LBP samplers. While the assumption used in [1] might be unnatural, this result is mainly used as a baseline comparison with LBP.
>
> **Assumptions for the target distributions**
>
> We agree that the current framework has a lot of assumptions and we agree that considering the impact of the gradient approximation is a very interesting topic. Investigations of this topic are ongoing.
>
> **Improper statements and typos**
>
> Thanks for pointing out the improper statements and typos. We have fixed them in the revision. Specifically, we have rephrased Lemma 3.3. For freezing tuning parameters, we actually followed [3] for tuning the step size $\epsilon$. After rereading the related sections in [3], we find that the method of freezing the tuning parameters is from [4].
>
> **Why use EJD as the metric?**
>
> Thanks for pointing out this gap in the presentation. In a continuous space, the limit converges to a diffusion process. As a result, the inverse autocorrelation only depends on the velocity of the diffusion process, which is independent of the function of interest [2]. In a discrete space, the limit converges to a jump process. As a result, the inverse autocorrelation only depends on the velocity of the jump process. Different from the diffusion process, whose velocity is characterized by ESJD, the velocity of a jump process is characterized by the EJD. In this case, we believe that the EJD is the best choice for the test functions. We have added the discussion about why selecting EJD in appendix B.1.
>
> **Algorithm box**
>
> We have edited Algorithm box 1 in main text to describe the ALBP algorithm.
>
> **Is the acceptance rate equal to the ratio of the normalizing constant Z in the general cases? Intuition for the results extrapolating well to more realistic scenarios.**
>
> Only for linear energy functions do the acceptance rates equal to the ratios of the normalizing constant Z. However,  in every M-H step, we only flip a small portion of the sites, where the gradient approximation gives a small error and the system is almost linear. Denote $\lambda(x)$ as the coefficient of the linear system associated with state $x$. The expected efficiency can be expressed as $\mathbb{E}[2l\Phi(-\frac{1}{2} \lambda(x) l^\frac{3}{2})] \approx 2l \Phi(-\frac{1}{2} \mathbb{E}[\lambda(x)] l^\frac{3}{2})$. When high probability states $x$ have similar coefficients $\lambda(x)$, the approximation holds. This is the intuition behind anticipating good extrapolation.
>
> **Why the ESS of RBM in table 1 is not consistent with EJD.**
>
> The reasons are twofold：
> * RBM is not a product distribution as assumed in the theorem. As a result, the ESS of a test function can be different from the ESS of EJD.
> * RBM trained on MNIST has a more complex modality than other models we evaluated. The test function we used is to project the 784 dimensional state to a scalar. Such a one dimensional projection does reflect the traverse of the original state. Although the ESS with respect to the projection is no longer proportional to the the ESS with respect to EJD, we still observe that they have strong positive correlation, which means that if the value of one variable increases when we change the target acceptance rate, the value of the other variable also increases.
>
> **Mathematical statement of Lemma 3.3**
>
> Thank the reviewer for pointing out our improper statement in Lemma 3.3. We have rephrased Lemma 3.3 from “sum … weakly converges to $A+B$” to “sum … - (A+B) converges to 0” in our revision.
>
> >[1] Gareth O Roberts. Optimal metropolis algorithms for product measures on the vertices of a hypercube. Stochastics and Stochastic Reports, 62(3-4):275–283, 1998. \
> [2] Gareth O Roberts and Jeffrey S Rosenthal. Optimal scaling for various metropolis-hastings algorithms. Statistical science, 16(4):351–367, 2001. \
> [3] Matthew D Hoffman, Andrew Gelman, et al. The no-u-turn sampler: adaptively setting path lengths in hamiltonian monte carlo. J. Mach. Learn. Res., 15(1):1593–1623, 2014 \
> [4] A. Gelman, J. Carlin, H. Stern, and D. Rubin. Bayesian Data Analysis. Chapman & Hall, 2004.

---

### Author Response · Authors · 2022-08-07
**Follow-Up**

Dear Reviewers,

Thanks again for your effort and valuable feedback during the review process. We have made our response and revised the paper accordingly. Since the deadline for reviewer-author discussion is approaching, we would like to kindly remind you to please let us know if your concerns have been resolved, or if you still have any questions that we can address during this period. Thank you so much for your understanding!

---

### Meta-Review · Area_Chair_bYdD · 2022-08-24

**Recommendation:** Accept
**Confidence:** Certain

**Metareview:**

The reviewers and I agree that the contributions of the paper are of interest and useful addition to the literature. Therefore, I recommend accepting the paper.

Please consider the reviewers' comments when preparing the camera-ready version.


**Award:**

No

---

### Decision · Program_Chairs · 2022-09-14

Accept